# Near-Optimal Dynamic Regret for Adversarial Linear Mixture MDPs

**Long-Fei Li, Peng Zhao, Zhi-Hua Zhou**

National Key Laboratory for Novel Software Technology, Nanjing University, China

School of Artificial Intelligence, Nanjing University, China

{lilf, zhaop, zhouzh}@lamda.nju.edu.cn

## Abstract

We study episodic linear mixture MDPs with the unknown transition and adversarial rewards under full-information feedback, employing *dynamic regret* as the performance measure. We start with in-depth analyses of the strengths and limitations of the two most popular methods: occupancy-measure-based and policy-based methods. We observe that while the occupancy-measure-based method is effective in addressing non-stationary environments, it encounters difficulties with the unknown transition. In contrast, the policy-based method can deal with the unknown transition effectively but faces challenges in handling non-stationary environments. Building on this, we propose a novel algorithm that combines the benefits of both methods. Specifically, it employs (i) an *occupancy-measure-based global optimization* with a two-layer structure to handle non-stationary environments; and (ii) a *policy-based variance-aware value-targeted regression* to tackle the unknown transition. We bridge these two parts by a novel conversion. Our algorithm enjoys an $\widetilde{\mathcal{O}}(d\sqrt{H^3K} + \sqrt{HK(H + \bar{P}_K)})$ dynamic regret, where $d$ is the feature dimension, $H$ is the episode length, $K$ is the number of episodes, $\bar{P}_K$ is the non-stationarity measure. We show it is minimax optimal up to logarithmic factors by establishing a matching lower bound. To the best of our knowledge, this is the *first* work that achieves *near-optimal* dynamic regret for adversarial linear mixture MDPs with the unknown transition without prior knowledge of the non-stationarity measure.

## 1 Introduction

Reinforcement Learning (RL) studies the problem where a learner interacts with the environments and aims to maximize the cumulative reward [Sutton and Barto, 2018], which has achieved significant success in games [Silver et al., 2016], robotics [Kober et al., 2013], large language model [Ouyang et al., 2022] and so on. The interaction is usually modeled as Markov Decision Processes (MDPs). Research on MDPs can be broadly divided into two lines based on the reward generation mechanism. The first line of work [Jaksch et al., 2010, Azar et al., 2013, 2017, He et al., 2021] considers the stochastic MDPs where the reward is sampled from a fixed distribution. In many real-world scenarios, however, the assumption of fixed reward distributions may not hold, as rewards can vary over time. This motivates the study on adversarial MDPs [Even-Dar et al., 2009, Yu et al., 2009, Zimin and Neu, 2013, Jin et al., 2020a], where rewards might change in an adversarial manner. To address the challenges of large-scale MDPs, recent studies have extended these two frameworks to incorporate function approximation, allowing RL algorithms to handle large state and action spaces. Two popular models are linear mixture MDPs [Ayoub et al., 2020] and linear MDPs [Jin et al., 2020b].

In this work, we focus on linear mixture MDPs with adversarial rewards, unknown transition and full-information feedback. Though significant advances have been achieved for this setting [Cai et al., 2020, He et al., 2022], they choose *static regret* as the performance measure, which benchmarks the

38th Conference on Neural Information Processing Systems (NeurIPS 2024).

Table 1: Comparisons of dynamic regret guarantees with previous works studying adversarial linear mixture MDPs with the unknown transition and full-information feedback. Here, $d$ is the feature mapping dimension, $H$ is the episode length, $K$ is the number of episodes, $P_K$ and $\bar{P}_K$ are two kinds of non-stationarity measure defined in (3) satisfying $\bar{P}_K \leq HP_K$ [Zhao et al., 2022, Lemma 6].

| Reference | Dynamic Regret | $P_K$ or $\bar{P}_K$ |
|:---:|:---:|:---:|
| Zhong et al. [2021] | $\widetilde{\mathcal{O}}\big(dH^{7/4}K^{3/4} + H^2K^{2/3}P_K^{1/3}\big)$ | Known |
| Li et al. [2023] | $\widetilde{\mathcal{O}}\big(d\sqrt{H^3K} + H^2\sqrt{(K+P_K)(1+P_K)}\big)$ | Known |
| Li et al. [2024b] | $\widetilde{\mathcal{O}}\big(dHS\sqrt{K} + \sqrt{HK(H+\bar{P}_K)}\big)$ | Unknown |
| Upper Bound (Theorem 1) | $\widetilde{\mathcal{O}}\big(d\sqrt{H^3K} + \sqrt{HK(H+\bar{P}_K)}\big)$ | Unknown |
| Lower Bound (Theorem 2) | $\Omega\big(d\sqrt{H^3K} + \sqrt{HK(H+\bar{P}_K)}\big)$ | / |

learner's policies $\pi_1, \ldots, \pi_K$ against the *best-fixed* policy in hindsight, namely,

$$\text{Reg}_K = \max_{\pi \in \Pi} \sum_{k=1}^{K} V_{k,1}^{\pi}(s_{k,1}) - \sum_{k=1}^{K} V_{k,1}^{\pi_k}(s_{k,1}), \tag{1}$$

where $V_{k,1}^{\pi}(s_{k,1})$ is the expected cumulative reward of policy $\pi$ starting from initial state $s_{k,1}$ at episode $k$ and $\Pi$ is the policy set. While static regret is a natural choice for online MDPs, the best-fixed policy may perform poorly when the rewards change adversarially. To this end, an enhanced measure called *dynamic regret* is proposed in the literature [Zhao et al., 2022, Li et al., 2023], which benchmarks the learner's policies against a sequence of *changing* policies. This measure is defined as

$$\text{D-Reg}_K(\pi_{1:K}^c) = \sum_{k=1}^{K} V_{k,1}^{\pi_k^c}(s_{k,1}) - \sum_{k=1}^{K} V_{k,1}^{\pi_k}(s_{k,1}), \tag{2}$$

where $\pi_1^c, \ldots, \pi_K^c$ is any sequence of policies in the policy set that can be chosen with complete foreknowledge of online reward functions. The dynamic regret in (2) is a stronger notation as it recovers the static regret in (1) directly by setting $\pi_{1:K}^c \in \arg\max_{\pi \in \Pi} \sum_{k=1}^{K} V_{k,1}^{\pi}(s_{k,1})$. An ideal dynamic regret bound should scale with a certain variation quantity of compared policies denoted by $P_K(\pi_1^c, \ldots, \pi_K^c)$ or $\bar{P}_K(\pi_1^c, \ldots, \pi_K^c)$ that can reflect the degree of environmental non-stationarity.

While the flexibility of dynamic regret makes it well-suited for adversarial settings, it also presents significant challenges. The dynamic regret of *tabular* MDPs with full-information feedback has been thoroughly studied by Zhao et al. [2022] and Li et al. [2024b], who achieved optimal dependence on $K$ and $\bar{P}_K$ for the known and unknown transition settings, respectively, *without* requiring prior knowledge of the non-stationarity measure. However, the dynamic regret of adversarial linear mixture MDPs is still understudied. With the prior knowledge of the non-stationarity measure, Zhong et al. [2021] proposed a policy optimization algorithm with the restart strategy [Zhao et al., 2020], achieving a result with suboptimal dependence in $H$, $K$ and $P_K$. Later, Li et al. [2023] significantly improved their results by designing an algorithm with the optimal dynamic regret in $K$ and $P_K$, though the dependence on $H$ remains suboptimal. For the more challenging scenarios where non-stationarity is *unknown*, Li et al. [2023] made an initial solution by introducing a two-layer policy optimization algorithm, albeit with an additional term in the dynamic regret involving the switching number of the best base-learner. To address this limitation, Li et al. [2024b] developed an occupancy-measure-based algorithm with two-layer structure that achieves optimal dynamic regret in $K$ and $\bar{P}_K$. However, their result incurs a *polynomial* dependence on the state space size $S$, which is statistically undesirable.

In this work, we propose an algorithm that achieves the *near-optimal* dynamic regret in $d$, $H$, $K$ and $\bar{P}_K$ simultaneously for adversarial linear mixture MDPs with the unknown transition, without prior knowledge of the non-stationarity measure. We begin with in-depth analyses of the strengths and limitations of two most popular methods: occupancy-measure-based and policy-based methods. We find that while the occupancy-measure-based method is effective in addressing the non-stationary environments, it encounter difficulties with the unknown transition. In contrast, the policy-based method can deal with the unknown transition effectively but faces challenges in handling non-stationary environments. To this end, we propose a novel algorithm that combines the benefits of both

methods. Specifically, our algorithm employs (i) an *occupancy-measure-based global optimization* with a two-layer framework to handle the non-stationary environments; and (ii) a *policy-based variance-aware value-targeted regression* to handle the unknown transition. We bridge these two parts through a novel conversion. We show our algorithm achieves an $\widetilde{\mathcal{O}}(d\sqrt{H^3 K} + \sqrt{HK(H + \bar{P}_K)})$ dynamic regret and prove it is minimax optimal up to logarithmic factors by establishing a matching lower bound. Table 1 presents the comparison between our result and previous works. Our result surpasses all previous results, even those that require prior knowledge of the non-stationarity measure.

We note a similar combination was firstly employed in Ji et al. [2024], but for distinctly different purposes. In their work, the occupancy-measure-based component is used to provide a horizon-free (independent of horizon length $H$) static regret whereas our objective is to address non-stationary environments. One limitation of this hybrid approach is the computational complexity, which is dominated by the occupancy-measure-based component and thus expensive compared to policy-based method. This issue is the inherent challenge for occupancy-measure-based method and also appears in several studies [Zhao et al., 2023, Ji et al., 2024]. Nevertheless, our analyses suggest that the occupancy-measure-based method offers unique advantages in handling non-stationary environments. Investigating whether similar results can be attained by other computationally efficient methods is an important future work. We believe our work represents a significant step forward, as it is reasonable to prioritize achieving *statistical optimality* before focusing on computational efficiency.

**Organization.** We review the related work in Section 2 and formulates the setup in Section 3. We analyze the challenges and introduce our algorithm in Section 4 and present the dynamic regret in Section 5. Section 6 concludes the paper. Due to the page limits, we defer all proofs to the appendices.

**Notations.** We denote by $[n]$ the set $\{1, \ldots, n\}$ and define $[x]_{[a,b]} = \min\{\max\{x, a\}, b\}$. For vector $x \in \mathbb{R}^d$ and positive semi-definite matrix $\Sigma \in \mathbb{R}^{d \times d}$, define $\|x\|_{\Sigma} = \sqrt{x^\top \Sigma x}$. For policies $\pi$ and $\pi'$, define $\|\pi - \pi'\|_{1,\infty} = \max_s \|\pi(\cdot|s) - \pi'(\cdot|s)\|_1$. The notation $\widetilde{\mathcal{O}}(\cdot)$ hides all polylogarithmic factors.

## 2 Related Work

In this section, we review related works on the dynamic regret of MDPs in non-stationary environments. The studies can be divided into two lines: non-stationary stochastic MDPs and non-stationary adversarial MDPs. These two categories address distinct challenges and are studied separately.

**Non-stationary Stochastic MDPs.** Non-stationary stochastic MDPs address scenarios where transitions and rewards are stochastically generated from *varying distributions*. The non-stationarity measure is typically defined as the total variation of the transitions or rewards over time. For infinite-horizon MDPs, the seminal work of Jaksch et al. [2010] investigates the piecewise stationary setting where both the transitions and rewards are subject to changes at specific time and remain fixed in between. Ortner et al. [2019] further advance the field by allowing for changes at every step. Subsequently, Cheung et al. [2020] introduce the Bandit-over-RL algorithm, which addresses the limitations of earlier works by eliminating the need for prior knowledge about the non-stationarity. Additional advancements have been made in episodic non-stationary MDPs [Mao et al., 2021, Domingues et al., 2021] and episodic non-stationary linear MDPs [Touati and Vincent, 2020, Zhou et al., 2022]. A breakthrough is the black-box method by Wei and Luo [2021], which can transform any algorithm with the optimal static regret under certain conditions, into another one that achieves optimal dynamic regret without prior knowledge of the non-stationarity. However, this method is *inapplicable* in adversarial settings. The limitation arises from its dependence on an optimistic estimator, constructed via a Upper Confidence Bound (UCB)-based algorithm for environmental change detection, a technique that performs well in stochastic environments but is less effective in adversarial scenarios.

**Non-stationary Adversarial MDPs.** Non-stationary adversarial MDPs consider settings where the rewards are generated in an *adversarial* manner. The non-stationarity measure is defined as the variation of *arbitrary* changing compared policies, allowing the policy to adapt to non-stationary environments implicitly. An illustrative difference between non-stationary stochastic and adversarial MDPs is that, in some cases, even if the rewards and transitions change over time, the optimal policy may remain fixed. The dynamic regret of *tabular* MDPs with full-information feedback has been thoroughly studied by Zhao et al. [2022] and Li et al. [2024b], who achieved optimal dependence on $K$ and $\bar{P}_K$ for the known and unknown transition settings respectively, *without* requiring prior knowledge of the non-stationarity measure. However, the dynamic regret of adversarial linear mixture

MDPs is still understudied. With the prior knowledge about the non-stationarity measure, Zhong et al. [2021] proposed a policy optimization algorithm with restart strategy [Zhao et al., 2020], achieving a result with suboptimal dependence in $H$, $K$ and $P_K$. Later, Li et al. [2023] significantly improved their results by designing an algorithm with optimal dynamic regret in terms of $K$ and $P_K$, though the dependence on $H$ remains suboptimal. For the more challenging scenarios where non-stationarity is *unknown*, Li et al. [2023] made an initial solution by introducing a two-layer policy optimization algorithm, albeit with an additional term in the dynamic regret involving the switching number of the best base-learner. To address this limitation, Li et al. [2024b] developed an occupancy-measure-based algorithm with two-layer structure [Zhang et al., 2018, Yan et al., 2023, Zhao et al., 2024] that achieves optimal dynamic regret in $K$ and $\bar{P}_K$. However, their dynamic regret incurs a *polynomial* dependence on the state space size $S$, which is undesirable. In this work, we propose an algorithm that achieves near-optimal dynamic regret in $d$, $H$, $K$ and $\bar{P}_K$ simultaneously for adversarial linear mixture MDPs with the unknown transition, without prior knowledge of the non-stationarity measure.

## 3 Problem Setup

We focus on episodic MDPs with the unknown transition and adversarial reward functions in the full-information feedback setting. We introduce the problem formulation in the following.

**Inhomogeneous, Episodic Adversarial MDPs.** We denote an inhomogeneous, episodic adversarial MDP by a tuple $\mathcal{M} = \{\mathcal{S}, \mathcal{A}, H, \{\mathbb{P}_h\}_{h \in [H]}, \{r_{k,h}\}_{k \in [K], h \in [H]}\}$, where $\mathcal{S}$ is the state space with cardinality $|\mathcal{S}| = S$, $\mathcal{A}$ is the action space with cardinality $|\mathcal{A}| = A$, $H$ is the length of each episode, $\mathbb{P}_h(\cdot|\cdot, \cdot) : \mathcal{S} \times \mathcal{A} \times \mathcal{S} \to [0, 1]$ is the transition with $\mathbb{P}_h(s'|s, a)$ denoting the probability of transiting to state $s'$ given the state $s$ and action $a$ at stage $h$, and $r_{k,h} : \mathcal{S} \times \mathcal{A} \to [0, 1]$ is the reward function for episode $k$ at stage $h$ chosen by the adversary. A policy $\pi = \{\pi_h\}_{h=1}^{H}$ is a collection of $h$ functions, where each $\pi_h : \mathcal{S} \to \Delta(\mathcal{A})$ maps a state $s$ to a distribution over action space $\mathcal{A}$. For any $(s, a) \in \mathcal{S} \times \mathcal{A}$, the state-action value function $Q_{k,h}^{\pi}(s, a)$ and value function $V_{k,h}^{\pi}(s)$ are defined as:

$$Q_{k,h}^{\pi}(s, a) = \mathbb{E}_{\pi}\left[\sum_{h'=h}^{H} r_{k,h'}(s_{h'}, a_{h'}) \,\Big|\, s_h = s, a_h = a\right], \quad V_{k,h}^{\pi}(s) = \mathbb{E}_{a \sim \pi_h(\cdot|s)}[Q_{k,h}^{\pi}(s, a)],$$

where the expectation is taken over the randomness of $\pi$ and $\mathbb{P}$. For any function $V : \mathcal{S} \to \mathbb{R}$, we define $[\mathbb{P}_h V](s, a) = \mathbb{E}_{s' \sim \mathbb{P}_h(\cdot|s,a)}[V(s')]$ and $[\mathbb{V}_h V](s) = [\mathbb{P}_h V^2](s, a) - ([\mathbb{P}_h V](s, a))^2$.

The interaction protocol is given as follows. At the beginning of episode $k$, the environment chooses the reward functions $\{r_{k,h}\}_{h \in [H]}$ and decides the initial state $s_{k,1}$, where the reward function may be chosen in an adversarial manner. Simultaneously, the learner decides a policy $\pi_k = \{\pi_{k,h}\}_{h \in [H]}$. Starting from the initial state $s_{k,1}$, the learner chooses an action $a_{k,h} \sim \pi_{k,h}(\cdot|s_{k,h})$, obtains the reward $r_{k,h}(s_{k,h}, a_{k,h})$, and transits to the next state $s_{k,h+1} \sim \mathbb{P}_h(\cdot|s_{k,h}, a_{k,h})$ for $h \in [H]$. After the episode $k$ ends, the learner observes the entire reward function $\{r_{k,h}\}_{h \in [H]}$. The goal of the learner is to minimize the dynamic regret in (2). Denote by $T = KH$ the total steps.

**Linear Mixture MDPs.** We focus on *linear mixture MDPs*, which was introduced by Ayoub et al. [2020] and has been studied by subsequent works [Cai et al., 2020, Zhou et al., 2021, Li et al., 2024c].

**Definition 1** (Linear Mixture MDPs). An MDP $M = \{\mathcal{S}, \mathcal{A}, H, \{\mathbb{P}_h\}_{h \in [H]}, \{r_{k,h}\}_{k \in [K], h \in [H]}\}$ is called an inhomogeneous, episode $B$-bounded linear mixture MDP, if there exist a *known* feature mapping $\phi(s'|s, a) : \mathcal{S} \times \mathcal{A} \times \mathcal{S} \to \mathbb{R}^d$ and an *unknown* vector $\theta_h^* \in \mathbb{R}^d$ with $\|\theta_h^*\|_2 \leq B, \forall h \in [H]$, such that (i) $\mathbb{P}_h(s'|s, a) = \phi(s'|s, a)^{\top} \theta_h^*$, (ii) $\|\phi_V(s, a)\|_2 \triangleq \|\sum_{s' \in \mathcal{S}} \phi(s'|s, a) V(s')\|_2 \leq 1$ for any $(s, a) \in \mathcal{S} \times \mathcal{A}$ and any bounded function $V : \mathcal{S} \to [0, 1]$.

**Occupancy Measure.** We introduce the concept of occupancy measure [Altman, 1998, Zimin and Neu, 2013]. Given a policy $\pi$ and a transition $\mathbb{P}$, the occupancy measure $q$ is defined as the probability of visiting state-action-state triple $(s, a, s')$ under transition $\mathbb{P}$ and policy $\pi$, that is,

$$q_h^{\mathbb{P}, \pi}(s, a, s') = \Pr[s_h = s, a_h = a, s_{h+1} = s' \mid \mathbb{P}, \pi].$$

A valid occupancy measure $q = \{q_h\}_{h=1}^{H}$ satisfies the following properties. First, each stage is visited exactly once and thus $\forall h \in [H]$, $\sum_{s \in \mathcal{S}} \sum_{a \in \mathcal{A}} \sum_{s' \in \mathcal{S}} q_h(s, a, s') = 1$. Second, the probability of entering a state when coming from the previous stage equals to the probability of leaving from that state to the next stage, i.e., $\forall s \in \mathcal{S}$, $\sum_{a \in \mathcal{A}} \sum_{s' \in \mathcal{S}} q_1(s, a, s') = \mathbb{1}\{s = s_1\}$ and $\forall h \in [2, H]$,

$\sum_{a \in \mathcal{A}} \sum_{s' \in \mathcal{S}} q_h(s, a, s') = \sum_{s'' \in \mathcal{S}} \sum_{a \in \mathcal{A}} q_{h-1}(s'', a, s)$. For any occupancy measure $q$ satisfying the above two properties, it induces a transition $\mathbb{P}^q = \{\mathbb{P}_h^q\}_{h=1}^H$ and a policy $\pi^q = \{\pi_h^q\}$ such that

$$\mathbb{P}_h^q(s'|s, a) = \frac{q_h(s, a, s')}{\sum_{s''} q_h(s, a, s'')}, \pi_h^q(a|s) = \frac{\sum_{s'} q_h(s, a, s')}{\sum_{a', s'} q_h(s, a', s')}, \forall (s, a, s', h) \in \mathcal{S} \times \mathcal{A} \times \mathcal{S} \times [H].$$

We denote by $\Delta$ the set of all occupancy measures satisfying the above two properties. For a transition $\mathbb{P}$, denote by $\Delta(\mathbb{P}) \in \Delta$ the set of occupancy measures whose induced transition $\mathbb{P}^q$ is exactly $\mathbb{P}$. For a collection of transitions $\mathcal{P}$, denote by $\Delta(\mathcal{P}) \in \Delta$ the set of occupancy measures whose induced transition $\mathbb{P}^q$ is in the transition set $\mathcal{P}$. We use $q_k = q^{\mathbb{P}, \pi_k}, q_k^c = q^{\mathbb{P}, \pi_k^c}$ to simplify the notation.

**Non-stationarity measure.** The non-stationarity measure aims to quantify the non-stationarity of the environments. We introduce two kinds of non-stationarity measures widely used in the literature:

$$P_K = \sum_{k=2}^K \sum_{h=1}^H \|\pi_{k,h}^c - \pi_{k-1,h}^c\|_{1,\infty}, \quad \bar{P}_K = \sum_{k=2}^K \sum_{h=1}^H \|q_{k,h}^c - q_{k-1,h}^c\|_1. \tag{3}$$

They quantify the difference between the compared policies and the compared occupancy measures, respectively. Zhao et al. [2022, Lemma 6] show it holds that $\bar{P}_K \leq H P_K$. Thus, we focus on the $\bar{P}_K$-type upper bound in this work as it implies an upper bound in terms of $H P_K$ directly.

# 4 The Proposed Algorithm

In this section, we first analyze the strengths and limitations of two most popular methods for adversarial MDPs. Then we propose a novel algorithm that combines the benefits of both approaches.

## 4.1 Analysis of Two Popular Methods

Occupancy-measure-based and policy-based methods are the two most popular approaches for solving adversarial MDPs. Both methods have been extensively studied in the literature and shown to enjoy favorable static regret guarantees [Zimin and Neu, 2013, Cai et al., 2020]. However, when it comes to dynamic regret, both methods face significant challenges. We introduce the details below.

### 4.1.1 Framework I: Occupancy-measure-based Method

The first line of work [Zimin and Neu, 2013, Rosenberg and Mansour, 2019, Jin et al., 2020a] employed the occupancy-measure-based method for adversarial MDPs. This method use the occupancy measure as a proxy for the policy, optimizing over the occupancy measure rather than the policy directly. While the value function is not convex in the policy space, it becomes convex in the occupancy measure space, making this approach theoretically more attractive compared to policy-based method. However, this shift introduces new challenges for dynamic regret analysis, as discussed below.

Using the concept of the occupancy measure, the dynamic regret in (2) can be rewritten as $\text{D-Reg}_K(\pi_{1:K}^c) = \sum_{k=1}^K V_{k,1}^{\pi_k^c}(s_{k,1}) - \sum_{k=1}^K V_{k,1}^{\pi_k}(s_{k,1}) = \sum_{k=1}^K \langle q_k^c - q_k, r_k \rangle$. By this conversion, the online MDP problem is reduced to the standard online linear optimization problem over the occupancy measure set $\Delta(\mathbb{P})$ induced by the true transition $\mathbb{P}$. However, the true transition $\mathbb{P}$ is unknown and thus the decision set $\Delta(\mathbb{P})$ is inaccessible. To address this issue, a general idea is to construct a confidence set $\mathcal{P}_k$ at episode $k$ that contains the true transition with high probability then replace the decision set by $\Delta(\mathcal{P}_k)$. Then the dynamic regret can be decomposed into:

$$\text{D-Reg}_K(\pi_{1:K}^c) = \sum_{k=1}^K \langle q_k^c - q_k, r_k \rangle = \underbrace{\sum_{k=1}^K \langle q_k^c - \hat{q}_k, r_k \rangle}_{\texttt{D-Regret-OLO}} + \underbrace{\sum_{k=1}^K \langle \hat{q}_k - q_k, r_k \rangle}_{\texttt{approximation-error}}, \tag{4}$$

where $\hat{q}_k$ is an occupancy measure in the decision set $\Delta(\mathcal{P}_k)$. The first term is the dynamic regret of online linear optimization over the decision set $\Delta(\mathcal{P}_k)$, which has been explored in the known transition setting [Zhao et al., 2022]. This term can be well controlled by a two-layer structure, as demonstrated in the online learning literature [Zhang et al., 2018, Yan et al., 2023, Zhao et al., 2024].

The second term reflects the approximation error introduced by using the confidence set $\mathcal{P}_k$ as a surrogate for the true transition $\mathbb{P}$, which constitutes the primary challenge of this approach.

**Key Difficulties.** To control the approximation error in (4), previous works [Rosenberg and Mansour, 2019, Jin et al., 2020a] proposed to bound the term $\sum_{k=1}^{K} \|\hat{q}_k - q_k\|_1$, leveraging Hölder's inequality: $\langle \hat{q}_k - q_k, r_k \rangle \leq \|\hat{q}_k - q_k\|_1 \|r_k\|_\infty$ and $r_k \in [0,1]^{SA}$. While this approach is effective for tabular MDPs, it fails to exploit the inherent structure of linear mixture MDPs. Although the transition kernel $\mathbb{P}$ exhibit a linear structure, the occupancy measure is not linear and retains a complex recursive form, as highlighted in Zhao et al. [2023]. Consequently, the regret bound resulting from this method depends on the state space size $S$, which is undesirable for linear mixture MDPs.

**Remark 1.** Occupancy-measure-based method optimizes a global object that encodes the entire policy across all states, which sacrifices some computational efficiency to better handle non-stationary environments. Thanks to their global optimization property, these methods offer favorable dynamic regret guarantees. However, they struggle to address the *unknown transition* in linear mixture MDPs and suffer from an undesirable dependence on the state space size $S$ in the dynamic regret bound.

### 4.1.2 Framework II: Policy-based Method

Policy-based method directly optimizes the policy, making it easier to implement and computationally more efficient than occupancy-measure-based method. More importantly, it shows advantages in handling unknown transitions, as it only requires estimating the value function, bypassing the need to estimate the transition kernel explicitly. However, due to their inherent *local-search* nature, this method faces challenges in adapting to non-stationary environments and providing favorable dynamic regret guarantees. We outline the key challenges below.

By the performance difference lemma [Kakade and Langford, 2002, Cai et al., 2020], the dynamic regret in (2) can be rewritten as the following formulation:

$$\text{D-Reg}_K(\pi_{1:K}^c) = \sum_{k=1}^{K} \sum_{h=1}^{H} \mathbb{E}_{\pi_k^c} \left[ \langle Q_{k,h}^{\pi_k}(s_h, \cdot), \pi_{k,h}^c(\cdot|s_h) - \pi_{k,h}(\cdot|s_h) \rangle \right], \tag{5}$$

where the expectation is taken over the *changing* policy sequence $\pi_1^c, \ldots, \pi_K^c$. For each state $s_h$, the term $\sum_{k=1}^{K} \sum_{h=1}^{H} \langle Q_{k,h}^{\pi}(s_h, \cdot), \pi_{k,h}^c(\cdot|s_h) - \pi_{k,h}(\cdot|s_h) \rangle$ is exactly the regret of a $A$-armed bandit problem, with $Q_{k,h}^{\pi_k}(s_h, \cdot)$ being the "reward vector". Thus, it indicates the dynamic regret of online MDPs can be written as the weighted average of MAB dynamic regret over all states, where the weight for each state is its (*unknown* and *time-varying*) probability of being visited by $\pi_1^c, \ldots, \pi_K^c$.

Since the transition is unknown, the policy-based method need to estimate the value function $Q_{k,h}^{\pi_k}$ at each state. By the definition of linear mixture MDPs in Definition 1, for any $V_{k,h}(\cdot)$, it holds that $[\mathbb{P}_h V_{k,h+1}](s,a) = \langle \phi_{V_{k,h+1}}(s,a), \theta_h^* \rangle$. Therefore, learning the underlying transition parameters $\theta_h^*$ can be regarded as solving a "linear bandit" problem where the context is $\phi_{V_{k,h+1}}(s_{k,h}, a_{k,h})$ and the noise is $V_{k,h+1}(s_{k,h}, a_{k,h}) - [\mathbb{P}_h V_{k,h+1}](s_{k,h}, a_{k,h})$. This observation enables the application of "linear bandits" techniques to learn the value function effectively for the policy-based method.

The key challenge for the policy-based method lies in handling the non-stationarity of environment. Even with accurate estimates of $Q_{k,h}^{\pi_k}$ at each state, optimizing the dynamic regret remains difficult. To optimize (5), Li et al. [2023] proposed running the Exp3 algorithm [Auer et al., 2002] at each state and employing the fixed-share technique [Herbster and Warmuth, 2001, Cesa-Bianchi et al., 2012], forcing uniform exploration to deal with non-stationary environments. They update policies by

$$\pi_{k,h}'(\cdot|s) \propto \pi_{k-1,h}(\cdot|s) \exp\left( \eta \cdot Q_{k-1,h}^{\pi_{k-1}}(s, \cdot) \right), \quad \pi_{k,h}(\cdot|s) = (1-\gamma)\pi_{k,h}'(\cdot|s) + \gamma\pi^u(\cdot|s),$$

where $\eta$ is the step size, $\pi^u(\cdot|s)$ is uniform distribution, and $\gamma$ is the fixed-share parameter. They show it ensures $\text{D-Reg}_K(\pi_{1:K}^c) \leq \widetilde{\mathcal{O}}(\eta K H^3 + H(1+P_K)/\eta)$. To achieve a favorable dynamic regret, the step size $\eta$ needs to be set as $\eta \approx \sqrt{(1+P_K)/(KH^2)}$, which is impractical as the non-stationarity measure $P_K$ is *unknown*. The standard approach to address this issue in the online learning literature is to adopt the online ensemble framework [Zhao et al., 2024]. However, this approach encounters challenges in online MDPs as the dynamic regret in (5) involves the expectation of changing policies, which does not appear in the standard online learning setting. We elaborate on this issue below.

Specifically, the standard procedure of the two-layer framework is as follows. First, we construct a step size pool $\mathcal{H} = \{\eta_1, \ldots, \eta_N\}$ ($N = \mathcal{O}(\log K)$) to discretize the range of the optimal step size.

Subsequently, multiple base-learners $\mathcal{B}_1, \ldots, \mathcal{B}_N$ are maintained, with each associated with a step size $\eta_i \in \mathcal{H}$. Finally, a meta-algorithm is employed to track the unknown best base-learner. Then, the dynamic regret can be decomposed into two parts: (i) the dynamic regret of the best base-learner; and (ii) the regret of the meta-algorithm to track the best-learner. Formally,

$$\underbrace{\sum_{k=1}^{K} \mathbb{E}_{\pi_k^c} \left[ \sum_{h=1}^{H} \langle Q_{k,h}^{\pi_k}(s_h, \cdot), \pi_{k,h}^c(\cdot|s_h) - \pi_{k,h}^i(\cdot|s_h) \rangle \right]}_{\texttt{base-regret}} + \underbrace{\sum_{k=1}^{K} \mathbb{E}_{\pi_k^c} \left[ \sum_{h=1}^{H} \langle Q_{k,h}^{\pi_k}(s_h, \cdot), \pi_{k,h}^i(\cdot|s_h) - \pi_{k,h}(\cdot|s_h) \rangle \right]}_{\texttt{meta-regret}}.$$

**Key Difficulties.** Though the base-regret above involves the expectation of changing policies, it can be effectively controlled as the decomposition holds for any base-learner $\mathcal{B}_i$, allowing us to select the one with the optimal step size for analysis. However, controlling the meta-regret remains challenging as it still involves the expectation of changing policies and the optimal tuning is hindered.

**Remark 2.** By solely optimizing the policy at the visited states, the computational complexity of the policy-based method is independent of the state number $S$, offering a notable advantage. However, due to the non-stationary environments, more specifically, *changing weights* of different states, only caring about the visited states without considering their importance is not enough to achieve favorable dynamic regret. Thus, the local-search property enhances computational efficiency but poses difficulties in handling non-stationary environments. This reveals a significant trade-off between *computational efficiency* and the ability to manage *non-stationary environments*.

## 4.2 Our Method: A Novel Combination

By the analysis in Section 4.1, we observe that the occupancy-measure-based method is proficient in addressing non-stationary environments but shows limited compatibility with unknown transitions. In contrast, the policy-based method can deal with unknown transitions efficiently but faces challenges in handling non-stationary environments. To this end, we propose a new algorithm named **O**ccupancy-measure-based **O**ptimization with **P**olicy-based **E**stimation (OOPE), which combines the benefits of both methods. At a high level, OOPE algorithm consists of two components: (i) an *occupancy-measure-based global optimization* with a two-layer framework to deal with the non-stationarity of environments; and (ii) a *policy-based value-targeted regression* to handle the unknown transition. We bridge the two components through a novel analysis that converts the occupancy-measure-based approximation error into the policy-based estimation error. We elaborate on the details below.

### 4.2.1 Occupancy-measure-based Global Optimization

The occupancy-measure-based optimization follows Li et al. [2024b], using online mirror descent for updating the occupancy measure and a two-layer structure to manage non-stationary environments.

We first construct a step size pool $\mathcal{H} = \{\eta_1, \ldots, \eta_N\}$ to discretize the range of the optimal step size, then maintain multiple base-learners $\mathcal{B}_1, \ldots, \mathcal{B}_N$, each of which is associated with a step size $\eta_i \in \mathcal{H}$. Finally, we use a meta-algorithm to track the best base-learner. At each episode $k$, we construct a confidence set $\mathcal{C}_k$ such that $\theta_h^* \in \mathcal{C}_{k,h}$ with high probability. The details of the construction of confidence set will be introduced later. Then, the base-learner $\mathcal{B}_i$ updates the occupancy measure by

$$\hat{q}_k^i = \underset{q \in \Delta(\mathcal{C}_k, \alpha)}{\arg\max} \; \eta_i \langle q, r_k \rangle - D_\psi(q \| \hat{q}_{k-1}^i), \tag{6}$$

where $D_\psi(q\|q') = \sum_{s,a,s'} q(s,a,s') \ln \frac{q(s,a,s')}{q'(s,a,s')}$ is the KL-divergence and the decision set is set as $\Delta(\mathcal{C}_k, \alpha) = \{q_h \in [\alpha, 1]^{S^2 A}, \forall h \in [H] \mid q \text{ satisfies constraints } (\mathbf{C1}) \text{ and } (\mathbf{C2})\}$ where

$(\mathbf{C1}) : \sum_{a,s'} q_1(s,a,s') = \mathbb{1}\{s = s_{k,1}\}, \forall s \in \mathcal{S}; \quad \sum_{a,s'} q_h(s,a,s') = \sum_{a,s'} q_{h-1}(s',a,s) \, \forall (s,h) \in \mathcal{S} \times [2,H];$

$(\mathbf{C2}) : \forall (s,a,h) \in \mathcal{S} \times \mathcal{A} \times [H], \exists \theta \in \mathcal{C}_{k,h}, \; \frac{q_h(s,a,\cdot)}{\sum_{s'} q_h(s,a,s')} = \langle \phi(\cdot \mid s,a), \theta \rangle.$

The meta-algorithm updates weights by

$$p_k^i \propto p_{k-1}^i \exp(\varepsilon \langle \hat{q}_{k-1}^i, r_{k-1} \rangle), \tag{7}$$

where $\varepsilon > 0$ is the learning rate, $\langle \hat{q}_{k-1}^i, r_{k-1} \rangle$ evaluates the performance of the base-learner $\mathcal{B}_i$ at episode $k-1$. The final occupancy measure is given by $\hat{q}_k = \sum_{i=1}^{N} p_k^i \hat{q}_k^i$ and the learner plays the policy $\pi^{\hat{q}_k}$. Algorithm 1 summarizes the details. We show it enjoys the following guarantee.

| **Algorithm 1** OOPE | **Algorithm 2** CompConfSet |
|---|---|
| **Input:** step size pool $\mathcal{H}$, learning rate $\varepsilon$, and clipping parameter $\alpha$. | **Input:** Policy $\pi_k$, trajectory $U_k$, and reward $r_k$. |
| 1: Set $\hat{q}^i_{1,h}(s,a,s') = 1/(SAS), \forall i \in [N]$. | 1: **for** $h = H, H-1, \cdots, 1$ **do** |
| 2: Set $p^i_1 = 1/N, \forall i \in [N]$. | 2:    Set $Q_{k,h}(\cdot,\cdot)$ and $V_{k,h}(\cdot)$ as in (9). |
| 3: **for** $k = 1$ to $K$ **do** | 3:    $\widehat{\Sigma}_{k+1,h} \leftarrow \widehat{\Sigma}_{k,h} + \bar{\sigma}^{-2}_{k,h}\phi_{k,h,0}\phi^\top_{k,h,0}$. |
| 4:    Receive $\hat{q}^i_k$ by (6) from $\mathcal{B}_i$ for $i \in [N]$. | 4:    $\widehat{b}_{k+1,h} \leftarrow \widehat{b}_{k,h} + \bar{\sigma}^{-2}_{k,h}\phi_{k,h,0}V_{k,h+1}(s_{k,h+1})$. |
| 5:    Receive $p_k$ from meta-algorithm by (7). | 5:    $\widetilde{\Sigma}_{k+1,h} \leftarrow \widetilde{\Sigma}_{k,h} + \phi_{k,h,1}\phi^\top_{k,h,1}$. |
| 6:    Compute $\hat{q}_k = \sum^N_{i=1}p^i_k\hat{q}^i_k$. | 6:    $\widetilde{b}_{k+1,h} \leftarrow \widetilde{b}_{k,h} + \phi_{k,h,1}V^2_{k,h+1}(s_{k,h+1})$. |
| 7:    Play policy $\pi_k = \pi^{\hat{q}_k}$. | 7:    $\widehat{\theta}_{k+1,h} \leftarrow \widehat{\Sigma}^{-1}_{k+1,h}\widehat{b}_{k+1,h}$. |
| 8:    Observe reward $r_k$ and trajectory $U_k$. | 8:    $\widetilde{\theta}_{k+1,h} \leftarrow \widetilde{\Sigma}^{-1}_{k+1,h}\widetilde{b}_{k+1,h}$. |
| 9:    $\mathcal{C}_{k+1} \leftarrow$ CompConfSet $(\pi_k, U_k, r_k)$. | 9:    Compute confidence set $\mathcal{C}_{k+1}$ by (10). |
| 10: **end for** | 10: **end for** |

**Lemma 1.** *Suppose $\theta^*_h \in \mathcal{C}_{k,h}, \forall k \in [K], h \in [H]$. Set the clipping parameter $\alpha = 1/T^2$, the step size pool as $\mathcal{H} = \{\eta_i = 2^{i-1}\sqrt{K^{-1}\log(S^2A/H)} \mid i \in [N]\}$, where $N = \lceil\frac{1}{2}\log(1+\frac{4K\log T}{\log(S^2A/H)})\rceil + 1$, and the learning rate $\varepsilon = \sqrt{(\log N)/(HT)}$. Algorithm 1 ensures the following guarantee:*

$$\sum^K_{k=1}\langle q^c_k - \hat{q}_k, r_k\rangle \leq \mathcal{O}\left(\sqrt{T(H\log(S^2A) + \bar{P}_K\log T)}\right).$$

**Remark 3.** By the occupancy-measure-based optimization with a two-layer structure, we can handle the first term in (4) well. It remains to bound the `approximation-error` term $\sum^K_{k=1}\langle\hat{q}_k - q_k, r_k\rangle$, which arises from employing the confidence set $\mathcal{C}_k$ as a surrogate for true transition parameter $\theta^*$.

**Implementation Details of Algorithm 1.** The main computation complexity arises from the online mirror descent step of (6) in Line 4. This step can be divided into into an unconstrained optimization problem and a projection problem. The unconstrained optimization problem can be solved by the closed-form solution and the main computational cost lies in the projection step. Ji et al. [2024] show that though such a projection can not be formulated as a linear program, they can be efficiently solved by the Dysktra's algorithm as the decision set is an intersection of convex sets of explicit linear or quadratic forms. We refer the readers to Appendix D of Ji et al. [2024] for more details.

### 4.2.2 Occupancy Measure to Policy Conversion

As discussed in Section 4.1.1, previous works [Rosenberg and Mansour, 2019, Jin et al., 2020a, Li et al., 2024b] propose to control the approximation error by bounding the term $\sum^K_{k=1}\|\hat{q}_k - q_k\|_1$. However, though the transition $\mathbb{P}$ admits a linear structure, the occupancy measure does not and retains a complex recursive form, which introduces an undesired dependence on the state number $S$ in the final regret. To take advantage of strength of policy-based method in integrating with linear function approximation, we propose to learn value functions as a whole instead of directly controlling the occupancy measure discrepancies. This strategy diverges from traditional methods that bound $\langle\hat{q}_k - q_k, r_k\rangle$ by the transition discrepancies $\sum^H_{h=1}\sum_{s,a}\|\mathbb{P}_h(\cdot|s,a) - \bar{\mathbb{P}}_{k,h}(\cdot|s,a)\|_1$, where $\bar{\mathbb{P}}_k$ is the estimated transition in episode $k$. Instead, we opt to constrain $\langle\hat{q}_k - q_k, r_k\rangle$ through the value difference, which effectively integrates reward information. We introduce the details below.

Denote by $\widehat{V}_{k,1}(s_{k,1}) = \sum_{h,s,a,s'}\hat{q}_{k,h}(s,a,s')r_{k,h}(s,a)$ the expected reward given the occupancy measure $\hat{q}_{k,h}$. Then, the approximation error can be rewritten as

$$\sum^K_{k=1}\langle\hat{q}_k - q_k, r_k\rangle = \underbrace{\sum^K_{k=1}\left(\widehat{V}_{k,1}(s_{k,1}) - V_{k,1}(s_{k,1})\right)}_{\texttt{occupancy-policy-gap}} + \underbrace{\sum^K_{k=1}\left(V_{k,1}(s_{k,1}) - V^{\pi_k}_{k,1}(s_{k,1})\right)}_{\texttt{estimation-error}}, \quad (8)$$

where $V_{k,1}$ is an intermediate value we define later. Our key idea is building an optimistic estimator $V_{k,1}$ to ensure the first term is non-positive while controlling the second term. This bypasses the need to bound occupancy measure discrepancies, allowing us to focus solely on the value estimation error.

### 4.2.3 Policy-based Value-targeted Regression

It remains to build the value function to ensure $\widehat{V}_{k,h} \le V_{k,h}$. A key observation is that the occupancy measure $\hat{q}_k$ induces a new MDP whose transition lies in the confidence set $\mathcal{C}_k$ with high probability. Thus, it suffices to ensure that $V_{k,h}$ is an overestimate of the true value function in the confidence set.

By the definition of linear mixture MDPs, for any $V_{k,h}(\cdot)$, it holds that $[\mathbb{P}_h V_{k,h+1}](s,a) = \langle \phi_{V_{k,h+1}}(s,a), \theta_h^* \rangle$. Thus, we compute the optimistic estimation as follows:

$$Q_{k,h}(\cdot,\cdot) = \Big[ r_{k,h}(\cdot,\cdot) + \max_{\theta \in \mathcal{C}_{k,h}} \langle \theta, \phi_{V_{k,h+1}}(\cdot,\cdot) \rangle \Big]_{[0,H]}, \quad V_{k,h}(\cdot) = \mathbb{E}_{a \sim \pi_{k,h}(\cdot|\cdot)}[Q_{k,h}(\cdot,a)]. \quad (9)$$

where $\mathcal{C}_{k,h}$ is the confidence set. We introduce the details of constructing the confidence set below.

Following recent advances in linear mixture MDPs [Zhou et al., 2021, He et al., 2022], we estimate the parameter $\theta_h^*$ by the *weighted ridge regression* to utilize the variance information. Denote by $\phi_{k,h,0} = \phi_{V_{k,h+1}}(s_{k,h}, a_{k,h})$ and $\phi_{k,h,1} = \phi_{V_{k,h+1}^2}(s_{k,h}, a_{k,h})$. We construct the estimator $\widehat{\theta}_{k,h}$ as

$$\widehat{\theta}_{k,h} = \arg\min_{\theta \in \mathbb{R}^d} \sum_{j=1}^{k-1} \frac{[\langle \phi_{j,h,0}, \theta \rangle - V_{j,h+1}(s_{j,h+1})]^2}{\bar{\sigma}_{j,h}^2} + \lambda \|\theta\|_2^2,$$

where $\bar{\sigma}_{j,h}^2$ is the upper confidence bound of the variance $[\mathbb{V}_h V_{j,h+1}](s_{j,h}, a_{j,h})$ and is set as $\bar{\sigma}_{k,h}^2 = \max\{H^2/d, [\bar{\mathbb{V}}_{k,h} V_{k,h+1}](s_{k,h}, a_{k,h}) + E_{k,h}\}$, where $[\bar{\mathbb{V}}_{k,h} V_{k,h+1}](s_{k,h}, a_{k,h})$ is an estimate for the variance of value function $V_{k,h+1}$ under the transition $\mathbb{P}_h(\cdot|s_k, a_k)$, and $E_{k,h}$ is the bonus term to guarantee the true variance $[\mathbb{V}_{k,h} V_{k,h+1}](s_{k,h}, a_{k,h})$ is upper bounded by $[\bar{\mathbb{V}}_{k,h} V_{k,h+1}](s_{k,h}, a_{k,h}) + E_{k,h}$ with high probability. By definition, we have $[\mathbb{V}_h V_{k,h+1}](s_{k,h}, a_{k,h}) = \langle \phi_{k,h,1}, \theta_h^* \rangle - [\langle \phi_{k,h,0}, \theta_h^* \rangle]^2$. Thus, we set $[\bar{\mathbb{V}}_{k,h} V_{k,h+1}](s_{k,h}, a_{k,h}) = [\langle \phi_{k,h,1}, \widetilde{\theta}_{k,h} \rangle]_{[0,H^2]} - [\langle \phi_{k,h,0}, \widehat{\theta}_{k,h} \rangle]_{[0,H]}^2$, where $\widetilde{\theta}_{k,h}$ is used to estimate the second-order moment and is constructed as:

$$\widetilde{\theta}_{k,h} = \arg\min_{\theta \in \mathbb{R}^d} \sum_{j=1}^{k-1} [\langle \phi_{j,h,1}, \theta \rangle - V_{j,h+1}^2(s_{j,h+1})]^2 + \lambda \|\theta\|_2^2.$$

The confidence set for the parameter $\theta_h^*$ is constructed as

$$\mathcal{C}_{k,h} = \big\{ \theta \mid \|\widehat{\Sigma}_{k,h}^{1/2}(\theta - \widehat{\theta}_{k,h})\|_2 \le \widehat{\beta}_k \big\}. \quad (10)$$

where $\widehat{\Sigma}_{k,h}$ is a covariance matrix, and $\widehat{\beta}_k$ is a radius of the confidence set.

The algorithm is summarized in Algorithm 2. We show the approximation error is bounded as below.

**Lemma 2.** *Set the parameters as follows:*

$$\widehat{\beta}_k = 8\sqrt{d \log(1 + k/\lambda) \log(4k^2 H/\delta)} + 4\sqrt{d} \log(4k^2 H/\delta) + \sqrt{\lambda} B.$$
$$\bar{\beta}_k = 8d\sqrt{\log(1 + k/\lambda) \log(4k^2 H/\delta)} + 4\sqrt{d} \log(4k^2 H/\delta) + \sqrt{\lambda} B,$$
$$\widetilde{\beta}_k = 8H^2 \sqrt{d \log(1 + kH^4/(d\lambda)) \log(4k^2 H/\delta)} + 4H^2 \log(4k^2 H/\delta) + \sqrt{\lambda} B.$$
$$E_{k,h} = \min\Big\{ H^2, 2H\bar{\beta}_k \big\|\widehat{\Sigma}_{k,h}^{-1/2} \phi_{k,h,0}\big\|_2 \Big\} + \min\Big\{ H^2, \widetilde{\beta}_k \big\|\widetilde{\Sigma}_{k,h}^{-1/2} \phi_{k,h,1}\big\|_2 \Big\}.$$

*Algorithm 2 ensures with probability at least $1 - \delta$, it holds that*

$$\sum_{k=1}^{K} \langle \hat{q}_k - q_k, r_k \rangle \le \widetilde{\mathcal{O}}\Big( \sqrt{d^2 H^3 K} + \sqrt{d H^4 K} \Big).$$

**Remark 4.** This bound can be further simplified as $\widetilde{\mathcal{O}}(\sqrt{d^2 H^3 K})$ when $d \ge H$, which is a mild assumption. Following previous work [Zhou et al., 2021], we only discuss the case $d \ge H$ below.

## 5 Theoretical Guarantees

In this section, we present the dynamic regret upper bound and the lower bound for this problem.

## 5.1 Dynamic Regret Upper Bound and Lower Bound

The dynamic regret upper bound of our algorithm is guaranteed by the following theorem.

**Theorem 1.** *Set the parameters as in Lemma 1 and Lemma 2. Algorithm 1 with Algorithm 2 as the subroutine ensures with probability at least $1 - \delta$, the dynamic regret is upper bounded by*

$$\text{D-Reg}_K(\pi_{1:K}^c) \leq \widetilde{\mathcal{O}}\Big(\sqrt{d^2 H^3 K} + \sqrt{HK(H + \bar{P}_K)}\Big).$$

**Remark 5.** Compared with the dynamic regret of $\widetilde{\mathcal{O}}\big(\sqrt{d^2 H^3 K} + H^2\sqrt{(K + P_K)(1 + P_K)}\big)$ for *known* non-stationarity measure cases in Li et al. [2023], our bound has a better dependence on $H$. Compared with the dynamic regret of $\widetilde{\mathcal{O}}\big(dHS\sqrt{K} + \sqrt{HK(H + \bar{P}_K)}\big)$ for *unknown* non-stationarity measure cases in Li et al. [2024b], our bound removes the dependence on the state number $S$.

Then, we establish the dynamic regret lower bound for this problem.

**Theorem 2.** *Suppose $B \geq 2$, $d \geq 4$, $H \geq 3$, $K \geq (d-1)^2 H/2$, for any algorithm and any constant $\Gamma \in [0, 2KH]$, there exists an adversarial inhomogeneous linear mixture MDP and a policy sequence $\pi_1^c, \ldots, \pi_K^c$ such that $\bar{P}_K \leq \Gamma$ and $\text{D-Reg}_K(\pi_{1:K}^c) \geq \Omega(\sqrt{d^2 H^3 K} + \sqrt{HK(H + \Gamma)})$.*

**Remark 6.** Combining Theorem 1 and Theorem 2, our algorithm achieves the minimax optimal dynamic regret in terms of $d$, $H$, $K$ and $\bar{P}_K$ simultaneously up to logarithmic factors.

## 5.2 Proof Overview

We provide the proof sketch of Theorem 1. The detailed proof can be found in the appendixes.

**Proof Sketch** (of Theorem 1). We can decompose the dynamic regret as the following four terms:

$$\text{D-Reg}_K(\pi_{1:K}^c) = \underbrace{\sum_{k=1}^{K}\langle q_k^c - \hat{q}_k^i, r_k\rangle}_{\texttt{base-regret}} + \underbrace{\sum_{k=1}^{K}\langle \hat{q}_k^i - \hat{q}_k, r_k\rangle}_{\texttt{meta-regret}} + \underbrace{\sum_{k=1}^{K}\Big(\widehat{V}_{k,1} - V_{k,1}\Big)}_{\texttt{occupancy-policy-gap}} + \underbrace{\sum_{k=1}^{K}\Big(V_{k,1} - V_{k,1}^{\pi_k}\Big)}_{\texttt{estimation-error}}.$$

• *base-regret.* By the analysis of OMD, it can be upper bounded by $\widetilde{\mathcal{O}}\big(\eta_i KH + (H + \bar{P}_K)/\eta_i\big)$. Choosing the base-learner with the best step size leads to an upper bound of $\widetilde{\mathcal{O}}(\sqrt{KH(H + \bar{P}_K)})$.

• *meta-regret.* This term is the *static regret* of the expert-tracking problem. As our meta-algorithm is the Hedge algorithm, by the standard analysis, this term can be upper bounded by $\widetilde{\mathcal{O}}(H\sqrt{K})$.

• *occupancy-policy-gap.* As the value functions are optimistic estimators over the confidence set $\mathcal{C}_k$, the value gap between the occupancy measure and the policy is guaranteed to be non-positive.

• *estimation-error.* Let $\epsilon_{k,h}^Q = Q_{k,h} - Q_{k,h}^{\pi_k}$ and $\epsilon_{k,h}^V = V_{k,h} - V_{k,h}^{\pi_k}$, then define policy noise $M_{k,h,1} = \mathbb{E}_{a\sim\pi_k(\cdot|s_{k,h})}[\epsilon_{k,h}^Q(s_{k,h}, a)] - \epsilon_{k,h}^Q(s_{k,h}, a_{k,h})$, transition noise $M_{k,h,2} = [\mathbb{P}_h(\epsilon_{k,h}^V)](s_{k,h}, a_{k,h}) - \epsilon_{k,h}^V(s_{k,h+1})$ and bonus $\iota_{k,h} = Q_{k,h} - (r_{k,h} + \mathbb{P}_h V_{k,h+1})$. The estimation error can be decomposed into transition noises, policy noises, and bonuses, i.e., $\sum_{k=1}^{K}\sum_{h=1}^{H}(M_{k,h,1} + M_{k,h,2} + \iota_{k,h})$. The transition and policy noises can be bounded by $\widetilde{\mathcal{O}}(\sqrt{KH^3})$ using Azuma-Hoeffding's inequalities. The bonus term can be bounded by $\widetilde{\mathcal{O}}(\sqrt{d^2 H^3 K} + \sqrt{dH^4 K})$ according to Lemma 2. ∎

# 6 Conclusion and Future Work

In this work, we study the dynamic regret of adversarial linear mixture MDPs with the unknown transition. We observe the occupancy-measure-based method is effective in addressing non-stationary environments but struggles with unknown transitions. In contrast, the policy-based method can deal with unknown transitions effectively but faces challenges in handling non-stationary environments. To this end, we propose a new algorithm that combines the benefits of both methods, achieving an $\widetilde{\mathcal{O}}(\sqrt{d^2 H^3 K} + \sqrt{HK(H + \bar{P}_K)})$ dynamic regret without prior knowledge of the non-stationarity measure. We show it is optimal up to logarithmic factors by establishing a matching lower bound.

Currently, we achieve this result by employing a hybrid method. Exploring whether similar results can be attained using computationally more efficient methods is an important future work. Furthermore, extending our results to other MDP classes, such as generalized linear function approximation [Wang et al., 2021] and multinomial logit function approximation [Li et al., 2024a], is an interesting direction.

## Acknowledgments

This research was supported by National Science and Technology Major Project (2022ZD0114800) and NSFC (62361146852, 61921006).

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

# A Proof of Lemma 1

*Proof.* Without loss of generality, we assume that all states are reachable with positive probability under the uniform policy $\pi^u(a|s) = 1/A, \forall s \in \mathcal{S}, a \in \mathcal{A}$. Otherwise, we can simply remove the unreachable states from the state space. Assume $K$ is large enough such that the occupancy measure of $q^{\mathbb{P},\pi^u} \in \Delta(\mathbb{P}, 1/T)$. We define $u_k = (1 - \frac{1}{T})q_k^c + \frac{1}{T}q^{\mathbb{P},\pi^u} \in \Delta(\mathbb{P}, 1/T^2)$. Then, we can decompose the first term as

$$
\sum_{k=1}^{K} \langle q_k^c - \hat{q}_k, r_k \rangle = \sum_{k=1}^{K} \langle q_k^c - u_k, r_k \rangle + \sum_{k=1}^{K} \langle u_k - \hat{q}_k, r_k \rangle
$$

$$
= \frac{1}{T} \sum_{k=1}^{K} \langle q_k^c - q^{\mathbb{P},\pi^u}, r_k \rangle + \sum_{k=1}^{K} \langle u_k - \hat{q}_k, r_k \rangle
$$

$$
\leq \frac{1}{T} \sum_{k=1}^{K} \|q_k^c - q^{\mathbb{P},\pi^u}\|_1 \|r_k\|_\infty + \sum_{k=1}^{K} \langle u_k - \hat{q}_k, r_k \rangle
$$

$$
\leq 2 + \sum_{k=1}^{K} \langle u_k - \hat{q}_k, r_k \rangle
$$

$$
= 2 + \underbrace{\sum_{k=1}^{K} \langle u_k - \hat{q}_k^i, r_k \rangle}_{\texttt{base-regret}} + \underbrace{\sum_{k=1}^{K} \langle \hat{q}_k^i - \hat{q}_k, r_k \rangle}_{\texttt{meta-regret}}, \tag{11}
$$

which the first inequality follows from Holder's inequality, the second holds by $\|q_k^c - q^{\mathbb{P},\pi^u}\|_1 \leq 2H$, and the last holds for any $i \in [N]$. Next, we bound `base-regret` and `meta-regret` separately.

**Upper bound of base-regret.** Since the true transition parameter $\theta^*$ is contained in the confidence set $\mathcal{C}_k$ by condition, we ensure that $u_k \in \Delta(\mathbb{P}, 1/T^2) \in \Delta(\mathcal{C}_k, 1/T^2)$. By the update rule of $\hat{q}_k^i$ in (6), we ensure that $\hat{q}_k^i \in \Delta(\mathcal{C}_k, 1/T^2), \forall i \in [K]$. The update rule in (6) can be rewritten as:

$$
\bar{q}_{k+1}^i = \arg\max_{q \in \mathbb{R}^{HSAS}} \eta_i \langle q, r_k \rangle - D_\psi(q \| \hat{q}_k^i), \qquad \hat{q}_{k+1}^i = \arg\min_{q \in \Delta(\mathcal{C}_k, \alpha)} D_\psi(q \| \bar{q}_{k+1}^i),
$$

or equivalently,

$$
\bar{q}_{k+1,h}^i(s, a, s) = \hat{q}_{k,h}^i(s, a, s) \exp(\eta_i r_k(s, a)), \qquad \hat{q}_{k+1}^i = \arg\min_{q \in \Delta(\mathcal{C}_k, \alpha)} D_\psi(q \| \bar{q}_{k+1}^i).
$$

By the three-point identity in Lemma 9, we have

$$
\eta_i \langle u_k - \hat{q}_k^i, r_k \rangle = D_\psi(u_k \| \hat{q}_k^i) - D_\psi(u_k \| \hat{q}_{k+1}^i) + D_\psi(\hat{q}_k^i \| \bar{q}_{k+1}^i)
$$

$$
\leq D_\psi(u_k \| \hat{q}_k^i) - D_\psi(u_k \| \hat{q}_{k+1}^i) + D_\psi(\hat{q}_k^i \| \bar{q}_{k+1}^i), \tag{12}
$$

where the inequality is due to the Pythagorean theorem.

For the last term of (12), summing over $k$, we have

$$
\sum_{k=1}^{K} D_\psi(\hat{q}_k^i \| \bar{q}_{k+1}^i)
$$

$$
= \sum_{k=1}^{K} \sum_{h=1}^{H} \sum_{s,a,s'} \hat{q}_k^i(s, a, s') \left( -\eta_i r_k(s, a) - 1 + \exp(\eta_i r_k(s, a)) \right)
$$

$$
\leq \eta_i^2 \sum_{k=1}^{K} \sum_{h=1}^{H} \sum_{s,a,s'} \hat{q}_k^i(s, a, s') r_k(s, a)^2
$$

$$
\leq \eta_i^2 HK, \tag{13}
$$

where the first inequality is due to the fact $e^x - x - 1 \leq x^2$ for $x \in [0, 1]$.

For the first two terms of (12), by the definition of KL-divergence, we have

$$\sum_{k=1}^{K} \left( D_\psi(u_k\|\hat{q}_k^i) - D_\psi(u_k\|\hat{q}_{k+1}^i) \right)$$

$$= D_\psi(u_1\|\hat{q}_1^i) + \sum_{k=2}^{K} \left( D_\psi(u_k\|\hat{q}_k^i) - D_\psi(u_{k-1}\|\hat{q}_k^i) \right)$$

$$= D_\psi(u_1\|\hat{q}_1^i) + \sum_{k=2}^{K}\sum_{h=1}^{H}\sum_{s,a,s'} \left( u_{k,h}(s,a,s') \log \frac{u_{k,h}(s,a,s')}{\hat{q}_{k,h}^i(s,a,s')} - u_{k-1,h}(s,a,s') \log \frac{u_{k-1,h}(s,a,s')}{\hat{q}_{k,h}^i(s,a,s')} \right)$$

$$= D_\psi(u_1\|\hat{q}_1^i) + \psi(u_K) - \psi(u_1) + \sum_{k=2}^{K}\sum_{h=1}^{H}\sum_{s,a,s'} (u_{k,h}(s,a,s') - u_{k-1,h}(s,a,s')) \log \frac{1}{\hat{q}_{k,h}^i(s,a,s')}$$

$$\leq \underbrace{D_\psi(u_1\|\hat{q}_1^i) + \psi(u_K) - \psi(u_1)}_{I_1} + 2\log T \underbrace{\sum_{k=2}^{K}\sum_{h=1}^{H}\sum_{s,a,s'} |u_{k,h}(s,a,s') - u_{k-1,h}(s,a,s')|}_{I_2}$$

where the inequality holds by $\hat{q}_k^i \in \Delta(\mathcal{P}_k, 1/T^2)$. It remains to bound $I_1$ and $I_2$ separately.

For term $I_1$, since $\hat{q}_1^i$ minimize $\psi$ over $\Delta(\mathcal{P}_1, 1/T^2)$, we have $\langle \nabla\psi(\hat{q}_1^i), u_1 - \hat{q}_1^i \rangle \geq 0$, thus,

$$I_1 \leq \psi(u_K) - \psi(\hat{q}_1^i) \leq \sum_{h=1}^{H}\sum_{s,a,s'} \hat{q}_{1,h}^i(s,a,s') \frac{1}{\hat{q}_{1,h}^i(s,a,s')} \leq H\log(S^2A). \qquad (14)$$

For term $I_2$, for any $(s,a)$, we have

$$\sum_{s'} |u_{k,h}(s,a,s') - u_{k-1,h}(s,a,s')| = \sum_{s'} |u_{k,h}(s,a)\mathbb{P}_h(s'|s,a) - u_{k-1,h}(s,a,s')\mathbb{P}_h(s'|s,a)|$$

$$= \sum_{s'} |u_{k,h}(s,a) - u_{k-1,h}(s,a)|\mathbb{P}_h(s'|s,a)$$

$$= |u_{k,h}(s,a) - u_{k-1,h}(s,a)|.$$

Thus, we have

$$I_2 = \sum_{k=2}^{K}\sum_{h=1}^{H}\sum_{s,a} |u_{k,h}(s,a) - u_{k-1,h}(s,a)| = (1-\frac{1}{T})\|q_k^c - q_{k-1}^c\|_1 = (1-\frac{1}{T})\bar{P}_K \leq \bar{P}_K.$$

$$(15)$$

Combining (12), (13), (14) and (15), we have

$$\texttt{base-regret} \leq \frac{1}{\eta_i} \sum_{k=1}^{K} \left( D_\psi(u_k\|\hat{q}_k^i) - D_\psi(u_k\|\hat{q}_{k+1}^i) + D_\psi(\hat{q}_k^i\|\bar{q}_{k+1}^i) \right)$$

$$\leq \eta_i T + \frac{1}{\eta_i}(H\log(S^2A) + 2\bar{P}_K\log T).$$

It is easy to verify that the optimal step size is $\eta^* = \sqrt{(H\log(S^2A) + 2\bar{P}_K\log T)/T}$. Since $0 \leq \bar{P}_K \leq 2T$, we ensure that

$$\sqrt{\frac{H\log(S^2A)}{T}} \leq \eta^* \leq \sqrt{\frac{H\log(S^2A) + 4T\log T}{T}}.$$

By the construction of the step size pool $\mathcal{H} = \{\eta_i = 2^{i-1}\sqrt{K^{-1}\log(S^2A)} \mid i \in [N]\}$ with $N = 1 + \lceil \frac{1}{2}\log(1 + \frac{4K\log T}{\log(S^2A)}) \rceil$, we know that the step size therein is monotonically increasing with

$$\eta_1 = \sqrt{\frac{H\log(S^2A)}{T}}, \eta_N \geq \sqrt{\frac{H\log(S^2A) + 4T\log T}{T}}.$$

Thus, we ensure there exists an index $i^*$ such that $\eta_{i^*} \le \eta^* \le 2\eta_{i^*} = \eta_{i^*+1}$. Since the decomposition in (11) holds for any $\mathcal{B}_i$, we choose $\mathcal{B}_{i^*}$ to analyze the regret bound. By the definition of $\eta^*$, we have

$$
\begin{aligned}
\texttt{base-regret} &\le \eta_{i^*} T + \frac{1}{\eta_{i^*}}(H \log(S^2 A) + 2\bar{P}_K \log T) \\
&\le \eta^* T + \frac{2}{\eta^*}(H \log(S^2 A) + 2\bar{P}_K \log T) \\
&= 3\sqrt{T(H \log(S^2 A) + 2\bar{P}_K \log T)},
\end{aligned} \tag{16}
$$

where the last equality holds by substituting $\eta^* = \sqrt{(H \log(S^2 A) + 2\bar{P}_K \log T)/T}$.

**Upper bound of meta-regret.** Denote by $h_k^i = \langle \hat{q}_k^i, r_k \rangle \in [0, H]$, we have

$$
\texttt{meta-regret} = \sum_{k=1}^{K} \langle \hat{q}_k^i - \sum_{i=1}^{N} p_k^i \hat{q}_k^i, r_k \rangle = \sum_{k=1}^{K} \langle e_i - p_k, h_k \rangle,
$$

where $e_i$ is the $i$-th standard basis vector in $\mathbb{R}^N$. This is a standard Prediction with Expert Advice (PEA) problem and our algorithm is the well-known Hedge algorithm [Freund and Schapire, 1997, Herbster and Warmuth, 1998]. By the standard analysis of Hedge [Cesa-Bianchi and Lugosi, 2006, Theorem 2.2], we have

$$
\texttt{meta-regret} \le \frac{\log N}{\varepsilon} + \varepsilon H^2 K = \sqrt{HT \log N}, \tag{17}
$$

where the equality holds by setting $\varepsilon = \sqrt{(\log N / HT)}$.

Combining (11), (16) and (17), we have

$$
\sum_{k=1}^{K} \langle q_k^c - \hat{q}_k, r_k \rangle \le 3\sqrt{T(H \log(S^2 A) + 2\bar{P}_K \log T)} + 2.
$$

This finishes the proof. ∎

# B Proof of Lemma 2

In this section, we first provide the main proof of Lemma 2, and then present the proofs of the auxiliary lemmas used in the main proof.

## B.1 Main Proof

*Proof.* To prove Lemma 2, we first introduce the following lemma which shows the true parameter $\theta_h^*$ is contained in the confidence set $\mathcal{C}_{k,h}$ with high probability by such configuration.

**Lemma 3** (Zhou et al. [2021, Lemma 5])**.** *Let $\mathcal{C}_{k,h}$ be defined in (10) and set parameters as in Lemma 2. Then, we have $\theta_h^* \in \mathcal{C}_{k,h}$ for all $h \in [H]$ and $k \in [K]$ with probability at least $1 - 3\delta$.*

Denote by $\mathcal{E}$ the event when Lemma 3 holds, then $\Pr(\mathcal{E}) \ge 1 - 3\delta$. We are ready to prove Lemma 2.

First, we can rewrite the term $\sum_{k=1}^{K} \langle \hat{q}_k - q_k, r_k \rangle$ as

$$
\begin{aligned}
\sum_{k=1}^{K} \langle \hat{q}_k - q_k, r_k \rangle &= \sum_{k=1}^{K} \widehat{V}_{k,1}(s_{k,1}) - V_{k,1}^{\pi_k}(s_{k,1}) \\
&= \underbrace{\sum_{k=1}^{K} \left( \widehat{V}_{k,1}(s_{k,1}) - V_{k,1}(s_{k,1}) \right)}_{\texttt{occupancy-policy-gap}} + \underbrace{\sum_{k=1}^{K} \left( V_{k,1}(s_{k,1}) - V_{k,1}^{\pi_k}(s_{k,1}) \right)}_{\texttt{estimation-error}},
\end{aligned} \tag{18}
$$

Next, we bound these two terms separately.

**Upper bound of occupancy-iteration-gap.** This term is the gap between the value function computed by the occupancy measure $\{\hat{q}_k\}_{k=1}^{K}$ and the optimistic value function computed by backward iteration defined in (9). Similar to Ji et al. [2024, Lemma 6.1], we show this term is non-positive.

**Lemma 4.** *For any episode $k \in [K]$, define $\widehat{V}_{k,1}(s_{k,1}) = \sum_{h,s,a,s'} \hat{q}_{k,h}(s,a,s') r_{k,h}(s,a)$ and $V_{k,1}(s_{k,1})$ the value function computed in* (9), *on the event $\mathcal{E}$, it holds that $\widehat{V}_{k,1}(s_{k,1}) \leq V_{k,1}(s_{k,1})$.*

**Upper bound of estimation-error.** First, we present the following lemma which shows this term can be decomposed into three major terms, transition noise, policy noise and the sum of bonuses.

**Lemma 5.** *For all $k \in [K], h \in [H]$, it holds that*

$$V_{k,h}(s_{k,h}) - V_{k,h}^{\pi_k}(s_{k,h}) = \sum_{h'=h}^{H} \left( M_{k,h',1} + M_{k,h',2} - \iota_{k,h'}(s_{k,h'}, a_{k,h'}) \right),$$

*with*

$$M_{k,h,1} = \mathbb{E}_{a \sim \pi_k(\cdot \mid s_{k,h})}[Q_{k,h}(s_{k,h}, a) - Q_{k,h}^{\pi_k}(s_{k,h}, a)] - (Q_{k,h}(s_{k,h}, a_{k,h}) - Q_{k,h}^{\pi_k}(s_{k,h}, a_{k,h})),$$
$$M_{k,h,2} = [\mathbb{P}_h(V_{k,h+1} - V_{k,h+1}^{\pi_k})](s_{k,h}, a_{k,h}) - (V_{k,h+1}(s_{k,h+1}) - V_{k,h+1}^{\pi_k}(s_{k,h+1}))$$
$$\iota_{k,h}(\cdot, \cdot) = Q_{k,h}(\cdot, \cdot) - (r_{k,h}(\cdot, \cdot) + \mathbb{P}_h V_{k,h+1}(\cdot, \cdot)).$$

Note $M_{k,h,1}$ is the noise from the stochastic policy and $M_{k,h,2}$ is the noise from the state transition, Let $M_{k,h} = M_{k,h,1} + M_{k,h,2}, \forall k \in [K], h \in [H]$, we define two following high probability events:

$$\mathcal{E}_1 = \left\{ \forall h \in [H], \sum_{k=1}^{K} \sum_{h'=h}^{H} M_{k,h'} \leq 4\sqrt{H^3 K \log \frac{H}{\delta}} \right\}, \mathcal{E}_2 = \left\{ \sum_{k=1}^{K} \sum_{h=1}^{H} M_{k,h,2} \leq \sqrt{8H^3 K \log \frac{1}{\delta}} \right\}.$$

According to the Azuma-Hoeffding inequality, we have $\Pr(\mathcal{E}_1) \geq 1 - \delta$ and $\Pr(\mathcal{E}_2) \geq 1 - \delta$. It remains to bound the model prediction error $\iota_{k,h}$.

Next, we show the prediction error depends on the width of the confidence set and the cumulative estimate variance by the following lemma.

**Lemma 6.** *Define prediction error $\iota_{k,h} = Q_{k,h} - (r_{k,h} + \mathbb{P}_h V_{k,h+1})$, on the event $\mathcal{E}$, it holds that*

$$\sum_{k=1}^{K} \sum_{h=1}^{H} \iota_{k,h}(s_{k,h}, a_{k,h}) \leq 2\widehat{\beta}_K \sqrt{\sum_{k=1}^{K} \sum_{h=1}^{H} \bar{\sigma}_{k,h}^2} \sqrt{2Hd \log(1 + K/\lambda)}.$$

Here, $\bar{\sigma}_{k,h}^2$ is the estimated variance, for the total true variance $\sum_{k=1}^{K} \sum_{h=1}^{H} [\mathbb{V}_h V_{h+1}^{\pi^k}](s_{k,h}, a_h^k)$, we introduce the high probability event $\mathcal{E}_3$:

$$\mathcal{E}_3 = \left\{ \sum_{k=1}^{K} \sum_{h=1}^{H} [\mathbb{V}_h V_{h+1}^{\pi^k}](s_{k,h}, a_h^k) \leq 3(HK + H^3 \log(1/\delta)) \right\}.$$

Lemma C.5 in Jin et al. [2018] suggests that $\Pr(\mathcal{E}_3) \geq 1 - \delta$. Based on the events $\mathcal{E} \cap \mathcal{E}_1 \cap \mathcal{E}_2 \cap \mathcal{E}_3$, we have the following lemma which bounds the estimated variance of the value function.

**Lemma 7** (He et al. [2022, Lemma 6.5]). *On the events $\mathcal{E} \cap \mathcal{E}_1 \cap \mathcal{E}_2 \cap \mathcal{E}_3$, it holds that*

$$\sum_{k=1}^{K} \sum_{h=1}^{H} \bar{\sigma}_{k,h}^2 \leq \frac{2H^3 K}{d} + 179H^2 K + (165d^3 H^4 + 2062d^2 H^5) \log^2 \left( \frac{4K^2 H}{\delta} \right) \log^2 \left( 1 + \frac{KH^4}{\lambda} \right).$$

Combining Lemma 5, Lemma 6 and Lemma 7, we can bound the estimation-error as follows.

**Lemma 8.** *On the events $\mathcal{E} \cap \mathcal{E}_1 \cap \mathcal{E}_2 \cap \mathcal{E}_3$, for any $h \in [H]$, it holds that*

$$\sum_{k=1}^{K} V_{k,h}(s_{k,h}) - \sum_{k=1}^{K} V_{k,h}^{\pi_k}(s_{k,h}) \leq \widetilde{\mathcal{O}}\big( \sqrt{dH^4 K} + \sqrt{d^2 H^3 K} \big).$$

Finally, we finish the proof of Lemma 2 by combining Lemma 4 and Lemma 8. ∎

## B.2 Proofs of Auxiliary Lemmas

In this section, we proof the auxiliary lemmas used in the proof of Lemma 2 in Appendix B.1.

### B.2.1 Proof of Lemma 4

*Proof.* The proof is similar to that of Ji et al. [2024, Lemma 6.1]. The only difference is that $\hat{q}_k$ is a weighted combination rather than a single occupancy measure in the decision set. For $k \in [K]$, the occupancy measure $\hat{q}_k$ is given by $\hat{q}_k = \sum_{i=1}^{N} p_k^i \hat{q}_k^i$. Since $\hat{q}_k^i \in \Delta(\mathcal{P}_k, \alpha)$, we ensure that $\hat{q}_k \in \Delta(\mathcal{P}_k, \alpha)$. Thus, for occupancy measure $\hat{q}_k$, there exist $\bar{\theta}$ such that

$$\widehat{\mathbb{P}}_{k,h}(s'|s,a) = \frac{q_{k,h}(s,a,s')}{\sum_{s'} q_{k,h}(s,a,s')} = \langle \phi(s'|s,a), \bar{\theta}_{k,h}(s,a)\rangle, \forall (s,a,h) \in \mathcal{S} \times \mathcal{A} \times [H].$$

It is easy to verify that the update rule $\widehat{V}_{k,1}(s_{k,1}) = \sum_{h,s,a,a'} \hat{q}_{k,h}(s,a,s') r_{k,h}(s,a)$ computed by the occupancy measure is the same as the following backward iteration:

$$\widehat{Q}_{k,h}(s,a) = r_h(s,a) + \langle \phi_{\widehat{V}_{k,h+1}}(s'|s,a), \bar{\theta}_{k,h}(s,a)\rangle,$$

$$\widehat{V}_{k,h}(s) = \mathbb{E}_{a\sim\pi_k(\cdot|s)}\widehat{Q}_{k,h}(s,a), \quad \widehat{V}_{k,H+1}(s) = 0.$$

Then, we can prove this lemma by induction. The conclusion trivially holds for $n = H + 1$. Suppose the statement holds for $n = h + 1$, we prove it for $n = h$. For any $(s,a) \in \mathcal{S} \times \mathcal{A}$, since $\widehat{Q}_{k,h}(s,a) \leq H$, so if $Q_{k,h}(s,a) = H$, then it holds directly. Otherwise, we have

$$Q_{k,h}(s,a) - \widehat{Q}_{k,h}(s,a)$$
$$= \langle \phi_{V_{k,h+1}}(s,a), \widehat{\theta}_{k,h}\rangle + \widehat{\beta}\|\phi_{V_{k,h+1}}(s,a)\|_{\widehat{\Sigma}_{k,h}^{-1}} - \langle \phi_{\widehat{V}_{k,h+1}}(s'|s,a), \bar{\theta}_{k,h}(s,a)\rangle$$
$$\geq \langle \phi_{V_{k,h+1}}(s,a), \widehat{\theta}_{k,h}\rangle + \widehat{\beta}\|\phi_{V_{k,h+1}}(s,a)\|_{\widehat{\Sigma}_{k,h}^{-1}} - \langle \phi_{V_{k,h+1}}(s'|s,a), \bar{\theta}_{k,h}(s,a)\rangle$$
$$= \langle \phi_{V_{k,h+1}}(s,a), \widehat{\theta}_{k,h} - \bar{\theta}_{k,h}(s,a)\rangle + \widehat{\beta}\|\phi_{V_{k,h+1}}(s,a)\|_{\widehat{\Sigma}_{k,h}^{-1}}$$
$$\geq \widehat{\beta}\|\phi_{V_{k,h+1}}(s,a)\|_{\widehat{\Sigma}_{k,h}^{-1}} - \|\widehat{\theta}_{k,h} - \bar{\theta}_{k,h}(s,a)\|_{\widehat{\Sigma}_{k,h}}\|\phi_{V_{k,h+1}}(s,a)\|_{\widehat{\Sigma}_{k,h}^{-1}}$$
$$\geq 0,$$

where the first inequality holds by the inductive hypothesis, the second holds due to Holder's inequality, and the last holds due to $\bar{\theta}_{k,h}(s,a) \in \mathcal{C}_{k,h}$. By induction, we finish the proof. ∎

### B.2.2 Proof of Lemma 5

*Proof.* By the definition $V_{k,h}(s_{k,h}) = \mathbb{E}_{a\sim\pi_k(\cdot|s_{k,h})}[Q_{k,h}(s_{k,h},a)]$, we have

$$V_{k,h}(s_{k,h}) - V_{k,h}^{\pi_k}(s_{k,h})$$
$$= \mathbb{E}_{a\sim\pi_k(\cdot|s_{k,h})}\big[Q_{k,h}(s_{k,h},a) - Q_{k,h}^{\pi_k}(s_{k,h},a)\big]$$
$$= \mathbb{E}_{a\sim\pi_k(\cdot|s_{k,h})}\big[Q_{k,h}(s_{k,h},a) - Q_{k,h}^{\pi_k}(s_{k,h},a)\big] - \big(Q_{k,h}(s_{k,h},a_{k,h}) - Q_{k,h}^{\pi_k}(s_{k,h},a_{k,h})\big)$$
$$\quad + \big(Q_{k,h}(s_{k,h},a_{k,h}) - Q_{k,h}^{\pi_k}(s_{k,h},a_{k,h})\big)$$
$$= \mathbb{E}_{a\sim\pi_k(\cdot|s_{k,h})}\big[Q_{k,h}(s_{k,h},a) - Q_{k,h}^{\pi_k}(s_{k,h},a)\big] - \big(Q_{k,h}(s_{k,h},a_{k,h}) - Q_{k,h}^{\pi_k}(s_{k,h},a_{k,h})\big)$$
$$\quad + [\mathbb{P}_h(V_{k,h+1} - V_{k,h+1}^{\pi_k})](s_{k,h},a_{k,h}) + \iota_{k,h}(s_{k,h},a_{k,h})$$
$$= \underbrace{\mathbb{E}_{a\sim\pi_k(\cdot|s_{k,h})}\big[Q_{k,h}(s_{k,h},a) - Q_{k,h}^{\pi_k}(s_{k,h},a)\big] - \big(Q_{k,h}(s_{k,h},a_{k,h}) - Q_{k,h}^{\pi_k}(s_{k,h},a_{k,h})\big)}_{\triangleq M_{k,h,1}}$$
$$\quad + \underbrace{[\mathbb{P}_h(V_{k,h+1} - V_{k,h+1}^{\pi_k})](s_{k,h},a_{k,h}) - \big(V_{k,h+1}(s_{k,h+1}) - V_{k,h+1}^{\pi_k}(s_{k,h+1})\big)}_{\triangleq M_{k,h,2}}$$
$$\quad + \big(V_{k,h+1}(s_{k,h+1}) - V_{k,h+1}^{\pi_k}(s_{k,h+1})\big) + \iota_{k,h}(s_{k,h},a_{k,h})$$

where the third equality holds by the fact $Q_{k,h} = r_{k,h} + \mathbb{P}_h V_{k,h+1} + \iota_{k,h}$ and $Q_{k,h}^{\pi_k} = r_{k,h} + \mathbb{P}_h V_{k,h+1}^{\pi_k}$. Summing up the above equation from $h$ to $H$ recursively finishes the proof. ∎

### B.2.3 Proof of Lemma 6

*Proof.* By the definition of $\iota_{k,h} = Q_{k,h} - (r_{k,h} + \mathbb{P}_h V_{k,h+1})$, we have

$$
\begin{aligned}
&\iota_{k,h}(s,a)\\
&= Q_{k,h}(s,a) - (r_{k,h} + \mathbb{P}_h V_{k,h+1})(s,a)\\
&= r_{k,h}(s,a) + \left\langle \widehat{\theta}_{k,h}, \phi_{V_{k,h+1}}(s,a)\right\rangle + \widehat{\beta}_k \left\|\widehat{\Sigma}_{k,h}^{-1/2}\phi_{V_{k,h+1}}(s,a)\right\|_2 - (r_{k,h} + \mathbb{P}_h V_{k,h+1})(s,a)\\
&= \left\langle \widehat{\theta}_{k,h} - \theta_h^*, \phi_{V_{k,h+1}}(s,a)\right\rangle + \widehat{\beta}_k \left\|\widehat{\Sigma}_{k,h}^{-1/2}\phi_{V_{k,h+1}}(s,a)\right\|_2\\
&\leq 2\widehat{\beta}_k \left\|\widehat{\Sigma}_{k,h}^{-1/2}\phi_{V_{k,h+1}}(s,a)\right\|_2,
\end{aligned}
$$

where the first inequality holds by the configuration of $Q_{k,h}$ in (9), the second inequality holds by the definition of linear mixture MDP such that $[\mathbb{P}_h V_{k,h+1}](s,a) = \langle \phi_{V_{k,h+1}}(s,a), \theta_h^*\rangle$ and the last inequality holds by the construction of the confidence set in Lemma 3.

Then, we have

$$
\begin{aligned}
&\sum_{k=1}^{K}\sum_{h=1}^{H} \iota_{k,h}(s_{k,h}, a_{k,h})\\
&\leq \sum_{k=1}^{K}\sum_{h=1}^{H} 2\min\{\widehat{\beta}_k \|\widehat{\Sigma}_{k,h}^{-1/2}\phi_{V_{k,h+1}}(s_{k,h}, a_{k,h})\|_2, H\}\\
&\leq \sum_{k=1}^{K}\sum_{h=1}^{H} 2\widehat{\beta}_k \bar{\sigma}_{k,h} \min\left\{\|\widehat{\Sigma}_{k,h}^{-1/2}\phi_{V_{k,h+1}}(s_{k,h}, a_{k,h})/\bar{\sigma}_{k,h}\|_2, 1\right\}\\
&\leq 2\widehat{\beta}_K \sqrt{\sum_{k=1}^{K}\sum_{h=1}^{H}\bar{\sigma}_{k,h}^2}\sqrt{\sum_{k=1}^{K}\sum_{h=1}^{H}\min\left\{\|\widehat{\Sigma}_{k,h}^{-1/2}\phi_{V_{k,h+1}}(s_{k,h}, a_{k,h})/\bar{\sigma}_{k,h}\|_2, 1\right\}}\\
&\leq 2\widehat{\beta}_K \sqrt{\sum_{k=1}^{K}\sum_{h=1}^{H}\bar{\sigma}_{k,h}^2}\sqrt{2Hd\log(1 + K/\lambda)},
\end{aligned}
$$

where the first inequality holds by $Q_{k,h} \in [0, H]$, the second holds by $2\widehat{\beta}_k \bar{\sigma}_{k,h} \geq \sqrt{d}H/\sqrt{d} = H$, the third inequality is by Cauchy-Schwarz inequality and the last inequality holds by the elliptical potential lemma in Lemma 10. This finishes the proof. ∎

### B.2.4 Proof of Lemma 8

*Proof.* The proof can be obtained by combining Lemma 5, 6 and 7. Specifically, we have

$$
\begin{aligned}
&\sum_{k=1}^{K} V_{k,h}(s_{k,h}) - \sum_{k=1}^{K} V_{k,h}^{\pi_k}(s_{k,h})\\
&\leq 2\widehat{\beta}_K \sqrt{\sum_{k=1}^{K}\sum_{h=1}^{H}\bar{\sigma}_{k,h}^2}\sqrt{2Hd\log(1 + K/\lambda)} + 4\sqrt{H^3 K \log \frac{H}{\delta}} + \sqrt{8H^3 K \log \frac{1}{\delta}}\\
&\leq \widetilde{\mathcal{O}}\left(\sqrt{dH^4 K} + \sqrt{d^2 H^3 K}\right).
\end{aligned}
$$

where the first inequality holds by Lemma 5 and Lemma 6 and the last inequality holds by Lemma 7. This finishes the proof. ∎

## C  Proof of Theorem 1

*Proof.* Combining Lemma 1 and Lemma 2, we have

$$
\text{D-Reg}_K(\pi_{1:K}^c) \leq \mathcal{O}\left(\sqrt{T(H\log(S^2 A) + \bar{P}_K \log T)}\right) + \widetilde{\mathcal{O}}\left(\sqrt{dH^4 K} + \sqrt{d^2 H^3 K}\right).
$$

This finishes the proof. ∎

# D  Proof of Theorem 2

*Proof.* Our proof is similar to that of Li et al. [2023, Theorem 4]. At a high level, we prove this lower bound by noting that optimizing the dynamic regret of linear mixture MDPs is harder than (i) optimizing the static regret of linear mixture MDPs with the unknown transition, (ii) optimizing the dynamic regret of linear mixture MDPs with the known transition. Thus, we can consider the lower bound of these two problems separately and combine them to obtain the lower bound of the dynamic regret of linear mixture MDPs with the unknown transition.

First, we consider the lower bound of the static regret of adversarial linear mixture MDPs with the unknown transition. From lower bound in He et al. [2022, Theorem 5.3], since the dynamic regret recovers the static regret by choosing the best-fixed policy, we have the following lower bound for dynamic regret in this case:

$$\text{D-Reg}_K(\pi_{1:K}^c) \geq \Omega(\sqrt{d^2 H^3 K}). \tag{19}$$

Then, we consider the lower bound of the dynamic regret of adversarial linear mixture MDPs with the known transition. Zimin and Neu [2013] show the lower bound of the static regret for adversarial episodic loop-free MDP with known transition is $\Omega(H\sqrt{K \log(SA)})$. We note that though our MDP model is different from the episodic loop-free MDP, we can treat our MDP model to the episodic loop-free MDP with an expanded state space $\mathcal{S}' = \mathcal{S} \times [H]$. Thus, $\Omega(H\sqrt{K \log(SA)})$ is also a lower bound of the static regret for the MDP in our work. We consider the following two cases:

**Case 1:** $\Gamma \leq 2H$. In this case, we can directly utilize the lower bound of static regret as a lower bound of dynamic regret, i.e.,

$$\text{D-Reg}_K(\pi_{1:K}^c) \geq \Omega(H\sqrt{K \log(SA)}). \tag{20}$$

**Case 2:** $\Gamma > 2H$. Without loss of generality, we assume $L = \lceil \Gamma/2H \rceil$ divides $K$ and split the whole episodes into $L$ pieces equally. Next, we construct a special policy sequence such that the policy sequence is fixed within each piece and only changes in the split point. Since the sequence changes at most $L - 1 \leq \Gamma/2H$ times and the occupancy measure difference at each change point is at most $2H$, the total path length in $K$ episodes does not exceed $\Gamma$. As a result, we have

$$\text{D-Reg}_K(\pi_{1:K}^c) \geq LH\sqrt{K/L \log(SA)} = H\sqrt{KL \log(SA)} \geq \Omega(\sqrt{KH\Gamma \log(SA)}). \tag{21}$$

Combining (20) and (21), we have the following lower bound for the dynamic regret of adversarial linear mixture MDPs with the known transition kernel,

$$\text{D-Reg}_K(\pi_{1:K}^c) \geq \Omega\big(\max\{H\sqrt{K \log(SA)}, \sqrt{KH\Gamma \log(SA)}\}\big)$$
$$\geq \Omega(\sqrt{KH(H + \Gamma) \log(SA)}), \tag{22}$$

where the last inequality holds by $\max\{a, b\} \geq (a + b)/2$.

Combining two lower bounds (19) and (22), we have the lower bound of the dynamic regret of adversarial linear mixture MDPs with the unknown transition kernel,

$$\text{D-Reg}_K(\pi_{1:K}^c) \geq \Omega\big(\max\{\sqrt{d^2 H^3 K}, \sqrt{KH(H + \Gamma) \log(SA)}\}\big)$$
$$\geq \Omega\big(\sqrt{d^2 H^3 K} + \sqrt{KH(H + \Gamma) \log(SA)}\big).$$

This finishes the proof. ∎

# E  Supporting Lemmas

In this section, we introduce the supporting lemmas used in the proofs.

**Lemma 9** (Three-point identity). *Let $\mathcal{X}$ be a closed and convex set. For any $x \in \mathcal{X}$ and $y, z \in \text{int } \mathcal{X}$, it holds that*

$$D_\psi(x, y) + D_\psi(y, z) - D_\psi(x, z) = \langle \nabla\psi(z) - \nabla\psi(y), x - y \rangle.$$

**Lemma 10** (Abbasi-Yadkori et al. [2011, Lemma 11]). *Let $\{\mathbf{x}_t\}_{t=1}^\infty$ be a sequence in $\mathbb{R}^d$ space, $\mathbf{V}_0 = \lambda\mathbf{I}$ and define $\mathbf{V}_t = \mathbf{V}_0 + \sum_{i=1}^t \mathbf{x}_i\mathbf{x}_i^\top$. If $\|\mathbf{x}_i\|_2 \leq L, \forall i \in \mathbb{Z}_+$, then for each $t \in \mathbb{Z}_+$,*

$$\sum_{i=1}^t \min\left\{1, \|\mathbf{x}_i\|_{\mathbf{V}_{i-1}^{-1}}\right\} \leq 2d \log\left(\frac{d\lambda + tL^2}{d\lambda}\right).$$

