# OpenReview forum: "Near-Optimal Dynamic Regret for Adversarial Linear Mixture MDPs"
_NeurIPS.cc/2024/Conference — NeurIPS 2024 poster_

### Official Review · Reviewer_pW8k · 2024-07-09

**Soundness:** 4
**Presentation:** 4
**Contribution:** 3
**Rating:** 7
**Confidence:** 3

**Summary:**

This paper studies the adversarial MDPs, which assume unknown (but stochastic) transition models and adversarial rewards. This paper adopts the linear mixture MDP setting, and they aim to study the dynamic regret, where the comparison policy are allowed to be changed along $K$ steps.

The algorithm this paper proposed is a combination of policy optimization with exponential weights and online mirror descent. In the meantime, the mirror descent constraint set is updated according to the value target regression framework. This algorithm adopts the merits of both policy optimization and online linear optimization, and achieves optimal rate of regret.

**Strengths:**

This paper is well written. The algorithm, propositions and theorems are presented clearly.

From technical side, the algorithm is computationally efficient and very simple. The regret of the algorithm matches the lower bound as well.

**Weaknesses:**

All techniques used in this paper are existed in previous works. Apart from this, I don't see significant drawbacks in this paper.

**Questions:**

I have the following questions for the authors:

1. Does the algorithm apply to other type of non-stationary measures, e.g. switching cost, etc?

2. The lower bound presented in this paper is a constructive lower bound. Can you obtain instance dependent lower bound (given any the transition model, to obtain a lower bound adapts to the transition model)?

**Limitations:**

Yes. The authors addressed all the limitations listed in the guidelines.

---

> ### Author Rebuttal · Authors · 2024-08-06
>
> Thanks for your appreciation of our work! We will address your questions below.
>
> ---
>
> **Q1:** "Does the algorithm apply to other types of non-stationary measures, e.g. switching cost, etc?"
>
> **A1:** Thanks for your helpful question. Our algorithm can also be adapted to other types of non-stationary measures, such as the generalized path-length described by Hall & Willett (2013). This measure is defined as $P_K^\prime = \sum_{k=2}^K \sum_{h=1}^H \| q_{k,h}^c - \Phi_{k,h}(q_{k-1,h}^c) \|$, where $\Phi_{k,h}(\cdot)$ is a known dynamic model. When $\Phi_{k,h}(\cdot)$ is an identity function, this measure reduces to the path-length we used.
>
> The advantage of this generalized path-length measure is that it allows us to incorporate prior knowledge of the environment's non-stationarity into the dynamic model. If the dynamic model can predict the environment perfectly, (e.g., $q_{k,h}^c = \Phi_{k,h}(q_{k-1,h}^c)$), the generalized path-length $P_K^\prime$ will be zero, much smaller than the traditional path-length $P_K$. This capability can significantly enhance the performance of our algorithm in environments where the non-stationarity can be accurately modeled.
>
> As we focus on the basic setting to present our main results more clearly, we did not include this extension in the current version. We will include a discussion of this extension in the revised version to highlight the flexibility and applicability of our algorithm to various non-stationary measures.
>
> ---
>
> **Q2:** "Can you obtain instance dependent lower bound?"
>
> **A2:** Thank you for your insightful questions. Obtaining an instance-dependent lower bound is indeed an interesting and challenging problem. To the best of our knowledge, there is no existing instance-dependent lower bound that directly depends on the transition kernel, even for static regret. The main difficulty lies in identifying a suitable quantity (such as max/min value, rank, etc.) of the transition kernel that accurately characterizes the problem's hardness.
>
> There are other types of instance-dependent lower bounds in the literature, such as those that depend on the optimal value function. However, obtaining such a lower bound is more difficult than deriving a worst-case lower bound, which is already a challenging and open problem. We believe that achieving an instance-optimal dynamic regret is an exciting direction for future research, and our work represents a significant step toward this goal.
>
> ---
>
> **Q3:** "All techniques used in this paper exist in previous works."
>
> **A3:** Indeed, the high-level algorithmic ideas in our work exist in previous works. However, previous efforts failed to achieve optimal dynamic regret due to the inherent limitations of two methodologies in dynamic regret analysis, as highlighted in Sec 3.1 and 3.2. It was only after realizing that the two methodologies could complement each other, as we demonstrate, that optimal dynamic regret could be achieved. One of our technical contributions lies in revealing this crucial yet underexplored connection between the two most popular methods. We believe this optimal result is interesting and important for the community. Moreover, our technique for exploring the connection of two methodologies could be useful for broader problems in RL theory.
>
> ---
> We hope our responses address your concerns. We are happy to provide further clarification if needed. Thanks!
>
> ---
>
> **References:**
>
> [1] Hall, E. C., & Willett, R. M., Dynamical models and tracking regret in online convex programming. In ICML'13.

---

> > ### Comment · Reviewer_pW8k · 2024-08-12
> >
> > Thank you very much for your response. I do not have further questions.

---

### Official Review · Reviewer_PUu1 · 2024-07-13

**Soundness:** 3
**Presentation:** 3
**Contribution:** 3
**Rating:** 6
**Confidence:** 3

**Summary:**

The paper explores reinforcement learning in linear mixture Markov Decision Processes (MDPs) with adversarial rewards and unknown transitions. It analyzes policy-based and occupancy-measure-based methods, identifying their strengths and weaknesses. The paper introduced an algorithm that merges both approaches to achieve near-optimal dynamic regret, which is the first work that achieves this.

**Strengths:**

1. The paper proposes a novel algorithm that combines the strengths of policy-based and occupancy-measure-based methods
2. According to the authors, the paper achieves near-optimal dynamic regret for adversarial linear mixture MDPs, and is the first work that does this.
3. The paper is presented nicely and compared to previous approaches

**Weaknesses:**

The paper lacks experimental validation, but could be reasonable given it is a theoretical paper. Intuitively it would not scale to complex high dimensional environments empirically if it were implemented into a practical algorithm.

**Questions:**

None.

**Limitations:**

The authors discussed limitations of the work.

---

> ### Author Rebuttal · Authors · 2024-08-06
>
> Thank you for your positive feedback. We take this opportunity to highlight the key contributions of our work.
>
> ---
>
> **Q1:** "The paper lacks experimental validation, but could be reasonable given it is a theoretical paper..."
>
> **A1:** The primary goal of this work is to advance the theoretical understanding of adversarial linear mixture MDPs. The optimal dynamic regret of adversarial linear mixture MDPs is a fundamental and open question in RL theory. We design an algorithm that achieves the optimal dynamic regret for the first time, along with a matching lower bound. This accomplishment is made possible by revealing a crucial yet underexplored connection between two popular methodologies. We believe that our work represents a significant step forward in this area and provides valuable insights for the community. Further experimental validation and implementation into practical algorithms is an interesting and important direction for future work, although it is beyond the scope of this paper.
>
> ---
>
> We are happy to provide clarification if you have any further questions. Thanks!

---

> > ### Comment · Reviewer_PUu1 · 2024-08-07
> >
> > I appreciate the authors' response, which is reasonable. I will keep my score unchanged.

---

### Official Review · Reviewer_QFuD · 2024-07-15

**Soundness:** 3
**Presentation:** 3
**Contribution:** 3
**Rating:** 5
**Confidence:** 4

**Summary:**

This work studies adversarial Linear Mixture MDPs, where the reward function can vary across different episodes, and aims to analyze the dynamic regret, where the baseline policy can also change across different episodes with respect to the dynamic environment. The authors propose a novel algorithm with a theoretical guarantee of the dynamic regret. In addition, the authors provide a hard-to-learn instance, which suggests that the regret guarantee cannot be improved by any algorithm.

**Strengths:**

1. This work provide a algorithm with theoretically guarantee for the dynamic regret with adversarial environment, which is strictly stronger guarantee than the previous static regret analysis.

2. The authors provide a hard-to-learn instance, which suggests that the proposed algorithm has already achieved a near-optimal regret guarantee.

3. This paper is well-written and easy to follow.

**Weaknesses:**

1. The proposed algorithm is computationally inefficient. Firstly, calculating the feature $\phi{k,h}$ requires evaluating the value function over all possible next states. Secondly, the performance of the occupancy-measure-based method relies on solving a global optimization problem over all state-action pairs. For methods with linear function approximation, such state/action spaces are usually huge and require high computational costs in the previous two steps. Though this is a common issue in the analysis of linear mixture MDPs and occupancy-measure-based methods, it will still affect the impact of the proposed algorithm.

2. There is a similar setting that also considers the dynamic regret guarantee (e.g., [1]) in a non-stationary environment. This non-stationary MDP setting measures the environment's dynamics by the difference between adjacent reward/transition functions. In contrast, this work measures the dynamics by the difference between adjacent policy/occupancy. It is better to make a comparison with this setting and previous results. Otherwise, it is challenging to compare fairly with previous work or evaluate the bound of the dynamic regret in this study.

[1] Nonstationary Reinforcement Learning with Linear Function Approximation

**Questions:**

1.For the regret guarantee, why does there exist a $\sqrt{HK \cdot H}$ term? It is directly dominated by the first term $d\sqrt{H^3K}$.

2. The baseline policy $\pi_k^c$ should be chosen with knowledge of the online reward function before episode $k$, , or it can depend on the future reward function until the end of episode $K$.

---

> ### Author Rebuttal · Authors · 2024-08-06
>
> Thank you for your constructive review. We will address your questions below.
>
> ---
>
> **Q1:** "Why does there exist a $\sqrt{HK \cdot H}$ term in the regret bound? It is directly dominated by the first term $d\sqrt{H^3K}$"
>
> **A1:** Thanks for your careful observation. It is indeed directly dominated by the first term $d\sqrt{H^3K}$. We present the regret bound in this form to highlight the key three terms for this problem: the first term $d\sqrt{H^3K}$ represents the regret incurred from dealing with the unknown transition kernel, the second term $\sqrt{HK \cdot H}$ corresponds to the static regret when the environment is stationary, and the third term $\sqrt{HK \cdot P_K}$ captures the regret due to the non-stationarity of the environment. We will provide more intuition in the revised version.
>
> ---
>
> **Q2**: "The chosen of policy $\pi_k^c$ ..."
>
> **A2:** Yes, we can choose $\pi_k^c$ *arbitrarily*, allowing it to depend on the future reward function until the end of episode $K$. Our algorithm does not require the knowledge of the compared policy $\pi_k^c$ and our results hold universally for any sequence of compared policies. This flexibility enables our measure to adapt automatically to the non-stationary environment.
>
> ---
>
> **Q3:** "The proposed algorithm is computationally inefficient..."
>
> **A3:** We note that no prior work has achieved optimal dynamic regret, even for computationally inefficient algorithms. Therefore, the most central problem in this area is determining the optimal statistical complexity and how to achieve it. We design an algorithm with optimal dynamic regret for the first time, along with a matching lower bound. This is achieved by revealing a crucial yet underexplored connection between the two most popular methodologies, which is crucial for our results and could be useful for broader problems in RL theory. As the reviewer noted, computational complexity is a common issue for algorithms in this area and thus beyond the scope of this work. Achieving both computational efficiency and statistical optimality is an important and challenging future work. Nevertheless, our work has already made a significant step in this direction.
>
> ---
>
> **Q4:** "There is a similar setting that also considers the dynamic regret guarantee (e.g., [1]) in a non-stationary environment."
>
> **A4:** Thanks for pointing out this work, which also addresses the non-stationarity issue in MDPs and is thus related to our research. We will make sure to add a discussion on the connection between our work and [1] in the next version. However, it's crucial to highlight that the setting and results of [1] is fundamentally different from ours:
> - They study non-stationary **stochastic** MDPs, where the reward is assumed to be stochastically generated by parametric models with parameters continuously drifting. For example, the reward function is $r_k(s, a) = \theta_k^* \phi(s, a)$, where $\theta_k^*$ is drifting over time.
> - In contrast, we study non-stationary **adversarial** MDPs, allowing rewards adversarially chosen and do not make any stochastic assumption over the rewards. The objective is to be competitive with a sequence of time-varying compared policies.
>
> The algorithmic approaches to handle non-stationarity are also significantly different. For stochastic MDPs, methods such as sliding windows, restarts, and weighting are used. For adversarial MDPs, we employ two-layer structures. The optimal dynamic regret results also differ: $O(B^{1/3} K^{2/3})$ for stochastic MDPs and $O(P_K^{1/2} K^{1/2})$ for adversarial MDPs.
>
> An illustrative example highlighting the difference is when the difference between adjacent rewards scales linearly with time (i.e., $B=\Theta(K)$) but the optimal policy remains the same ($P_K = 0$). In this case, [1] suffers linear dynamic regret, while our dynamic regret remains sublinear.
>
> To summarize, the two settings and respective algorithms/results are incomparable. They can be viewed as two distinct models for non-stationary online MDPs.
>
> ---
>
> We hope our responses address your concerns. If your concerns have been addressed, we appreciate it if you could consider re-evaluating our work.

---

> > ### Author Response · Authors · 2024-08-13
> > **Thanks for the review! Have we properly addressed your concerns?**
> >
> > We sincerely appreciate your constructive feedback and are especially grateful for bringing the paper [1] to our attention. We will revise the paper to cite [1] and incorporate the above discussions in the next version.
> >
> > Given that the author-reviewer discussion period is soon coming to an end, please let us know if our response has properly addressed the concerns. We will be happy to provide clarification if you have any further questions. Thanks!
> >
> > Best,
> >
> > Authors

---

### Official Review · Reviewer_xu5Q · 2024-07-15

**Soundness:** 3
**Presentation:** 2
**Contribution:** 2
**Rating:** 6
**Confidence:** 1

**Summary:**

Disclaimer: This specific area falls outside my expertise, as indicated by my confidence score. Nevertheless, I have carefully reviewed this paper and the relevant literature to offer the most informed feedback possible.

This paper studies the dynamic regret for adversarial linear mixture MDPs, with unknown transitions and full-information feedback. It introduces a hybrid algorithm that combines a policy-based variance-aware value target regression method [Zhou 2021] with the occupancy-measure based method [Li et. al. 2024a]. The authors provide the dynamic regret analysis of this hybrid algorithm, demonstrating its near-optimality up to logarithmic factors by showing both upper and lower bound. In particular, it removes dependence on the state number S from the prior result in Li et. al 2024a.

**Strengths:**

It establishes a near-optimal dynamic regret for the first-time in its setting, improving over the results presented in [Li et al. 2024a],

**Weaknesses:**

1. I feel that the contributions of this paper could be highlighted better, given that the proposed algorithm involves various components from the literature. I’m not sure if I fully understand, but it seems that the main contribution lies more in the regret analysis rather than the algorithm itself. The proposed algorithm seems to be a combination of existing components from prior works [Li et. al 2024a and Zhou et al. 2021]. In particular, would it be fair to describe the proposed algorithm as Algorithm 1 in Li et. al 2024a, leveraging the techniques from UCRL-VTR+ [Zhou et. al 2021] to compute the confidence set, in place of the EstimateTransition routine in Algorithm 4 of Li et. al 2024a?
2. The organization and the clarity of Section 3 can be improved to help with better understanding. Although Section 3.1 and 3.2 seem to aim to outline the pros and cons to motivate the proposed hybrid approach in Section 3.3, a more concise presentation could improve readability. Additionally, reordering 3.2 and 3.1 may better align with the flow in Sec. 3.3.
3. The limitation of the work should explicitly mention the assumptions on access to the oracle in Algorithm 2. It’s a strong assumption and seems crucial in the ability to remove the dependence on the number of states S in the upper bound.

Minor: typos noted below

Line 243: The formula for KL-divergence

Line 326: “Combinin”

**Questions:**

In the weakness section, I noted my interpretation of the proposed algorithm. Could you elaborate if any new techniques are developed in this work? It will help me better understand the contributions of the work.

Can you comment on the intuition of why it is possible to remove the dependence on S in the upper bound?

**Limitations:**

The introduction acknowledges the limitation of lower computational efficiency due to the occupancy measure based component. But I think that the additional assumptions regarding access to the oracle, introduced by algorithm 2, should be explicitly stated as a limitation.

---

> ### Author Rebuttal · Authors · 2024-08-06
>
> Thanks for your helpful comments. We will address your questions below.
>
> ---
>
> **Q1:** "It seems that the main contribution lies more in the regret analysis rather than the algorithm itself. Could you elaborate if any new techniques are developed in this work?"
>
> **A1:** Thanks for your question. Both the algorithm and the regret analysis are new and non-trivial. Even though one may feel the high-level algorithmic ideas are similar to previous works, *previous efforts failed to achieve optimal dynamic regret* due to the inherent limitations of two methodologies in dynamic regret analysis, as highlighted in Sec 3.1 and 3.2. Our primary technical contribution lies in **revealing a crucial yet underexplored connection between the two most popular methods**: the occupancy-measure-based approach and the policy-based approach. These two methods are widely used in the literature, but they are typically considered separately. So It was **only** after realizing that the two methodologies could complement each other, as we demonstrate, that the new algorithm and optimal dynamic regret could be achieved. We believe this optimal result is interesting and important for the community. Moreover, our technique for exploring the connection of two methodologies is novel and could be useful for broader problems in RL theory. We will emphasize this point more clearly in the revised version.
>
> ---
>
> **Q2:** "Can you comment on the intuition of why it is possible to remove the dependence on $S$ in the upper bound?"
>
> **A2:** The key insight is that Li et al. (2024a) focus on the difference in occupancy measures (state-action distributions) between two policies, whereas we concentrate on the difference in their value functions (expected rewards). The value function difference can be much smaller than the difference between state-action distributions (e.g., when the rewards are all zeros, the value function differences are zero while the difference between state-action distributions can be arbitrary). This approach allows us to remove the dependence on $S$ in the upper bound. We will provide more intuition in the revised version.
>
> ---
>
> **Q3:** "Would it be fair to describe the proposed algorithm as .."
>
> **A3:** The main component of our algorithm is the occupancy-measure-based global optimization (Algorithm 1 in Li et. al 2024a) and the policy-based value-targeted regression (UCRL-VTR+ [Zhou et. al 2021]). However, the key insight is to combine these two methodologies in a novel way to achieve optimal dynamic regret. We connect the two methodologies by using the occupancy measure to policy conversion in Section 3.3.2, which is non-trivial and requires careful analysis.
>
> ---
>
> **Q4:** "The organization and the clarity of Section 3 can be improved to help with better understanding."
>
> **A4:** Thanks for the suggestion. We will reorder 3.2 and 3.1 to better align with the flow in Sec. 3.3 and present Section 3 more concisely.
>
> ---
>
> **Q5:** "The limitation of the work should explicitly mention the assumptions on access to the oracle in Algorithm 2 ..."
>
> **A5:** We appreciate your observation. This assumption is not the reason for removing the dependence on $S$ in the upper bound. In fact, this is a standard assumption in the literature and has been used in **all** existing works on linear mixture MDPs (e.g., Zhou et al., 2021, He et al., 2022, Li et al., 2023). This assumption is just used to compute the $Q$-function by backward induction. The Oracle can be estimated by Monte Carlo methods in practice. We will clarify this point in the revised version.
>
> ---
>
> We hope our responses clarify the technical contributions and address your concerns. If your concerns have been addressed, we appreciate it if you could consider re-evaluating our work.
>
> **References:**
>
> [1] Zhou, D., Gu, Q., & Szepesvari, C., Nearly Minimax Optimal Reinforcement Learning for Linear Mixture Markov Decision Processes. In COLT'21.
>
> [2] He J., Zhou D., & Gu Q., Near-optimal Policy Optimization Algorithms for Learning Adversarial Linear Mixture MDPs. In AISTATS'22.
>
> [3] Li, L. F., Zhao, P., & Zhou, Z. H., Dynamic Regret of Adversarial Linear Mixture MDPs. In NeurIPS'23.

---

> > ### Comment · Reviewer_xu5Q · 2024-08-11
> >
> > Thank you for providing the clarifications. I have increased the score accordingly.

---

> > > ### Author Response · Authors · 2024-08-12
> > > **Thank you!**
> > >
> > > Thank you for your re-evaluation and the revised score. We are happy to discuss work with you.

---

### Decision · Program_Chairs · 2024-09-25

**Decision:**

Accept (poster)

**Comment:**

In the challenging setting of linear-mixture MDPs with adversarial rewards and unknown transitions, this paper presents a new algorithm that combines the advantages of policy-based and occupancy-based approaches. The authors prove that the algorithm is near optimal, the first such result for this setting.

The reviewers agree that the improvement over the state of the art is significant and that the paper is well written. Several points were clarified in the discussion phase, most importantly: the importance of the algorithmic contribution, the role of the assumptions, the main proof idea that led to the theoretical improvement, computational efficiency, and differences w.r.t. previous work studying dynamic regret in a similar setting.

In recommending acceptance, I stress the importance of incorporating the clarifications provided to reviewers in the camera ready version.